# Modeling genome-wide enzyme evolution predicts strong epistasis underlying catalytic turnover rates

David Heckmann[1], Daniel C. Zielinski[1] & Bernhard O. Palsson [1,2]

Systems biology describes cellular phenotypes as properties that emerge from the complex interactions of individual system components. Little is known about how these interactions have affected the evolution of metabolic enzymes. Here, we combine genome-scale metabolic modeling with population genetics models to simulate the evolution of enzyme turnover numbers ($k_{cat}$s) from a theoretical ancestor with inefficient enzymes. This systems view of biochemical evolution reveals strong epistatic interactions between metabolic genes that shape evolutionary trajectories and influence the magnitude of evolved $k_{cat}$s. Diminishing returns epistasis prevents enzymes from developing higher $k_{cat}$s in all reactions and keeps the organism far from the potential fitness optimum. Multifunctional enzymes cause synergistic epistasis that slows down adaptation. The resulting fitness landscape allows $k_{cat}$ evolution to be convergent. Predicted $k_{cat}$ parameters show a significant correlation with experimental data, validating our modeling approach. Our analysis reveals how evolutionary forces shape modern $k_{cat}$s and the whole of metabolism.

---

[1] Department of Bioengineering, University of California, San Diego, La Jolla, CA 92093-0412, USA. [2] The Novo Nordisk Foundation Center for Biosustainability, Technical University of Denmark, 2800 Lyngby, Denmark. Correspondence and requests for materials should be addressed to B.O.P. (email: palsson@ucsd.edu)

The biological systems we observe today are the results of evolutionary trajectories that were shaped by their underlying genotype-to-fitness map, termed the fitness landscape. The components of the system constantly change to increase fitness in the current environment. It is thus tempting to assume that, given the right environment, biological systems can be described as the state that results in the highest fitness possible under all biophysical constraints. Whereas such optimality assumptions were successfully applied to understand a variety of systems properties like bacterial growth rates[1,2], gene expression patterns[3–5], and metabolic fluxes[6,7], they are expected to prove futile when the underlying fitness landscape is rugged and exhibits local optima[8,9], or when the natural selection cannot overcome genetic drift to establish potential fitness gains[10–12]. The topography of the fitness landscape is determined by epistasis[13], i.e., the extent to which the fitness effect of a mutation depends on the genetic background. Understanding epistasis is thus crucial for understanding evolutionary dynamics and constraints, and systems models can serve as a key tool to understand these interactions[9,14,15].

It was suggested that the catalytic turnover numbers ($k_{cat}$s) of metabolic enzymes constitute an example of a system state that is distant from a potential optimum, as the efficiency of most enzymes remains far from its theoretical maximum[16,17]. Enzyme turnover numbers span over six orders of magnitude and are essential for understanding biological processes on a quantitative level, as they quantitatively describe the proteomic demands of reaction flux, growth, and thus fitness[2,18–23]. In contrast to this high variability and functional importance, experimental data on $k_{cat}$ is scarce (data in the enzyme kinetics database BRENDA[24] accounts for about 10% of the reactions in the E. coli model iJO1366[16,25]) and exhibits high noise[16]. An improved understanding of the evolutionary and biophysical forces that shape the distribution of kinetic parameters on a systems scale would thus constitute an important step towards quantitative understanding of cellular metabolism. A meta-analysis of databases of $k_{cat}$s showed two major patterns[16]. On the one hand, $k_{cat}$s in primary metabolism are consistently higher than those in pathways of secondary metabolism, a finding that can be interpreted as the result of differential selection pressure on the respective genes. On the other hand, the underlying biochemical mechanism has a measurable effect on $k_{cat}$, suggesting that an interplay between biophysical and evolutionary constraints determines metabolic $k_{cat}$s. How these factors have acted mechanistically to result in the diverse kinetic turnover numbers we observe today is unknown.

The study of evolution is often limited to retrospective phylogenetic analysis of genome sequences. Nevertheless, when the selective advantage conferred by a metabolic system can be identified, quantitative models can be used to predict fitness correlates and evolution. In the past, systems models of metabolism have been used successfully to describe a variety of evolutionary phenomena like the dynamics of genome reduction[26], properties of ancient metabolism[27], the global optimum of metabolic adaptation[1], and the trajectories of photosynthesis evolution[28]. In this study, we aim to understand the evolutionary mechanisms that underlie $k_{cat}$ evolution and its apparent failure to reach optimality. As $k_{cat}$s provide a quantitative link between proteome costs and metabolic flux, metabolic models can be used to predict how $k_{cat}$s affect growth as a proxy for fitness. To this end, we combine genome-scale modeling of metabolism with population genetics models to simulate how modern $k_{cat}$s evolved from slow ancestors in a network context. We predict that $k_{cat}$ evolution is convergent and constrained by strong epistasis. In order to validate the model, we compare end points of our evolutionary simulations to experimental turnover rates from in vitro and in vivo sources.

## Results

**A model for simulating systems-wide $k_{cat}$ evolution.** As $k_{cat}$s affect fitness by controlling the proteomic cost of enzyme reactions[2,18,19,29], we hypothesize that genome-scale models of cell growth can be used to retrace $k_{cat}$ evolution in a network context.

The core structure of the metabolic network is conserved across the tree of life[30,31], and thus modern metabolic networks can be expected to contain information about the network context in which enzymes evolved. Because of the quality of its metabolic reconstruction and the relatively high coverage of kinetic data, we choose the metabolic network of E. coli K-12 MG1655 as an ideal candidate to study $k_{cat}$ evolution.

To predict $k_{cat}$-dependent growth as a proxy for fitness, we use the MOMENT algorithm[4] and a genome-scale reconstruction of E. coli metabolism[25]. The MOMENT algorithm optimizes growth under a constraint on the total metabolic proteome a cell can sustain. As changes in gene expression can be achieved by the gene regulatory network of the cell or through mutations in a genetic target that is much larger than that for kinetic parameter evolution, we model gene expression as growth optimal.

Modern enzymes exhibit relatively high substrate specificity, but are assumed to have evolved from slow multifunctional ancestors[32–34]. We aim to model adaptation of kinetic turnover numbers after specificity increased, but where turnover numbers were still low. We thus assign turnover numbers of $10^{-3}$ $s^{-1}$, similar to the slowest enzymes observed today[16]. Starting from these ancestral slow enzymes, mutations are drawn randomly as multiplicative changes in $k_{cat}$s of a random reaction, where the majority are assumed to be decreasing $k_{cat}$ (decreasing:increasing = 100:1, see Fig. 1 and Methods). Whether a novel mutation achieves fixation is then calculated for the estimated effective population size of E. coli ($N_e = 2.5\text{e}7$[35,36]), and $k_{cat}$ evolution is simulated with a Markov Chain Monte Carlo approach (MCMC, Fig. 1). The model thus uses a strong-selection-weak-mutation regime[37].

As biological catalysts are limited to natural amino acids to stabilize transition states, it is expected that many reactions will have a distinct biophysical upper limit to the turnover rate that is lower than the theoretical limit resulting from diffusion rate of collisions. As certain reaction mechanisms were shown to consistently exhibit high $k_{cat}$s[16], we use the enzyme commission (EC) number to decide on a candidate set of 569 biophysically unconstrained reactions (see Methods). The remaining 1087 enzymes were considered biophysically constrained and were fixed to the median of in vitro $k_{cat}$ measurements (13.7 $s^{-1}$). In the context of evolutionary predictions, the number of enzymes in the constrained and unconstrained set are more meaningfully compared in terms of reactions that are contributing to growth. Flux variability analysis[38] for aerobic growth on glucose reveals that 278 growth-relevant reactions (see Methods) are unconstrained, while 183 carry a biophysical constraint; the majority of in silico growth-relevant reactions is thus evolving without upper limit.

**Evolutionary trajectories exhibit jumps and convergence.** When simulating $k_{cat}$ evolution with the MCMC algorithm, we can trace the dynamics of adaptation through evolutionary trajectories of growth rates (Fig. 2a). As a starting point, we choose an aerobic glucose environment. Ancestral slow enzymes cause initial growth rates to be low, but fixation of mutations that increase selected $k_{cat}$s leads to an irregular increase in growth rates that eventually saturates towards a growth rate close to 0.5 $h^{-1}$. This behavior is reproducible across replicates, and final growth rates are convergent across these independent evolutionary

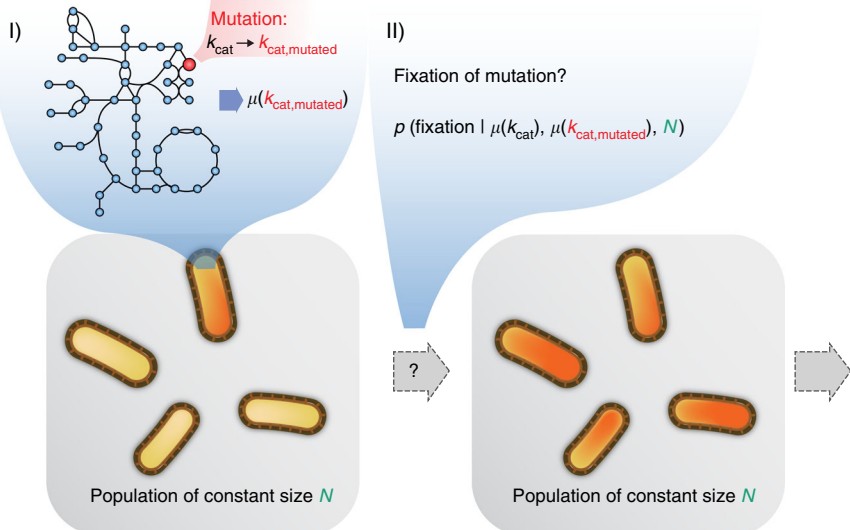

**Fig. 1** The MCMC algorithm used for simulating genome-scale $k_{cat}$ evolution. A single iteration of the algorithm proceeds as follows: (I) A mutation in the $k_{cat}$ of a random reaction of a single cell in the population is introduced. The original growth rate $\mu(k_{cat})$ and the novel growth rate $\mu(k_{cat,mutated})$ are predicted by solving the respective MOMENT problems (see Methods). (II) The probability of fixation for the novel mutation is calculated with a population genetics model based on $\mu(k_{cat})$, $\mu(k_{cat,mutated})$, and the population size $N$. Fixation of the novel change in $k_{cat}$ is then decided based on this probability. If fixation fails, the mutation is discarded. A typical simulation run includes around $10^8$ of the described iterations

trajectories. The average trajectory shows a sigmoidal shape that can be explained by a simple analytical model (Supplementary Note 1, Supplementary Figs. 16, 17, and 20), where variance in fitness is highest in intermediate states. Even though the majority of growth-contributing reactions—as determined by flux variability analysis[38]—were not assigned biophysical constraints on the evolution of higher $k_{cat}$s, growth rates are unable to surpass $0.5\,h^{-1}$, even when simulations are continued further than shown in Fig. 2 (Supplementary Figs. 2 and 5). This effect is the result of diminishing returns epistasis (DRE) acting between the evolving genes: the same mutation will result in a smaller fitness gain when the genetic background already enables a high growth rate (inset of Fig. 2b). Due to this effect, even large improvements in $k_{cat}$s of high-flux pathways can only confer a fitness benefit that approaches that of a neutral mutation and thus become subject to drift rather than selection[10,11] (Fig. 2b). We confirm this idea by using a greedy search that iteratively fixes the most beneficial mutations that double kcat: the maximum achievable fitness gain will reach the neutral barrier (where $s$ is smaller $\sim 1/N_e$[10,11]) without achieving a growth rate $>0.5\,h^{-1}$ (Supplementary Fig. 5). The underlying mechanism for the observed DRE is the dispersion of biophysical constraints through the shared metabolic proteome (Supplementary Note 1, Supplementary Fig. 4); as genome-wide adaptation progresses, improvements of already high $k_{cat}$s free up little protein that can be invested in limited reactions. This effect is independent of whether multiplicative or additive mutations are used and is particularly strong because many enzymes contribute to fitness (Supplementary Note 1, Supplementary Figs. 17, 18, 19, and 21). We simulated a maximum growth rate that ignores evolutionary constraints by setting the $k_{cat}$ of all unconstrained reactions to a value similar to the fastest known enzymes of $1e5\,s^{-1}$. We find a theoretically achievable growth rate of $1.58\,h^{-1}$, more than three times the rate of the evolved result. This result indicates the strong effect that DRE has in constraining $k_{cat}$ evolution: it acts to keep the system far from a theoretical fitness optimum.

Although in vitro data and biochemical reaction mechanisms defined our set of biophysically constrained reactions, the true identity of this set is unknown. We thus conducted a sensitivity

analysis for the identity and size of this set. The identity of the evolving set affects the final growth rate, but not the qualitative dynamics of adaptation or the occurrence of DRE (Supplementary Fig. 7). The speed of adaptation decreases with the size of the evolving set, as more reactions are required to acquire mutations to reach higher growth rates. An additional source of uncertainty comes from the nature of the distribution of mutational effects, which is unknown. We varied the mean of the distribution of mutational effects, but again found no effect on the qualitative dynamics of adaptation or the occurrence of DRE, but a small quantitative effect on the final growth rate (Supplementary Fig. 9).

**Multifunctional enzymes cause evolutionary jump dynamics.** In order to understand the irregular increase in growth rate observed in adaptive trajectories (Fig. 2A), we summarized genes for which mutation coincided with unusually high fitness gains. We found a small set of genes that was repeatedly associated with large jumps in fitness (Supplementary Table 1). When removing reactions catalyzed by the product of these genes, fitness jumps are drastically reduced and the speed of adaptation increases ($p < 2e-3$, Wilcoxon rank-sum test on the number of mutations required to reach half the end point growth rate), showing that they are indeed responsible for the irregular adaptation dynamics. Investigation of metabolic network model and gene-protein-reaction context of these genes revealed that all of them are multifunctional enzymes that catalyze multiple reactions in the same linear pathways. These genes are involved in histidine biosynthesis (histb), purine biosynthesis (purH), cell wall biosynthesis (glmU), and fatty acid biosynthesis (fabG). The irregular behavior in adaptive trajectories thus has a mechanistic reason that lies in the structure of the underlying network: protein cost of the linear pathway cannot be mitigated by increasing an individual $k_{cat}$ of a single active site, resulting in a fitness landscape that shows synergistic epistasis (Fig. 3). The pathway can then become a bottleneck for the adaptation process, where fixation of a specific neutral mutation in a multifunctional enzyme is required for further fitness gains (Fig. 3b).

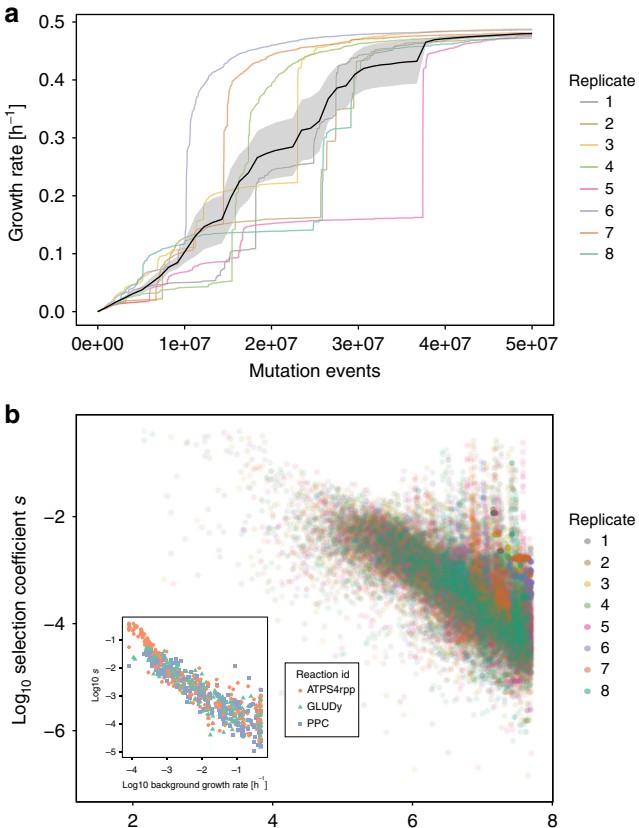

**Fig. 2** Evolutionary trajectories exhibit convergence and diminishing returns epistasis. **a** The growth rate of the population against the number of simulated mutations. The black line shows the average growth rate across replicates and its standard error. The replicates showed 4880 fixation events on average. See Supplementary Figure 3 for a fit of the analytical model presented in Supplementary Note 1. **b** The selection coefficient $s$ (defined as the change in growth rate relative to the novel growth rate) plotted against the cumulative number of simulated mutational events for all fixed mutations. The inset shows $s$ against the background growth rate in which a mutation occurred for the three reactions that had the most changes in $k_{cat}$ fixed. These reactions are: ATPS4rpp: ATP synthase (orange), GLUDy: Glutamate dehydrogenase (NADP) (green), and PPC: Phosphoenolpyruvate carboxylase (blue)

**Most reactions show repeatable evolution.** The high level of convergence that is exhibited in the adapted growth rates (Fig. 2a) is reflected in the turnover numbers of the evolved populations: vectors of adapted $k_{cat}$s show a high correlation across replicates (all Pearson's $R >= 0.9$, Supplementary Fig. 1). Clustering of the most divergent reactions reveals that the remaining differences in evolved $k_{cat}$s cannot be exclusively attributed to the stochasticity of the adaptation process: redundant metabolic routes in central carbon metabolism and redox metabolism cause $k_{cat}$ evolution to be divergent (Supplementary Fig. 1B). Nevertheless, $k_{cat}$ evolution is highly convergent and repeatable, indicating that similar patterns in turnover numbers across species could be the result of independent evolutionary trajectories.

**The evolved $k_{cat}$s agree with in vivo and in vitro data.** How well do our simulated end points of $k_{cat}$ evolution agree with experimental data on modern $k_{cat}$s? In order to answer this question, we simulate $k_{cat}$ evolution in randomly changing model

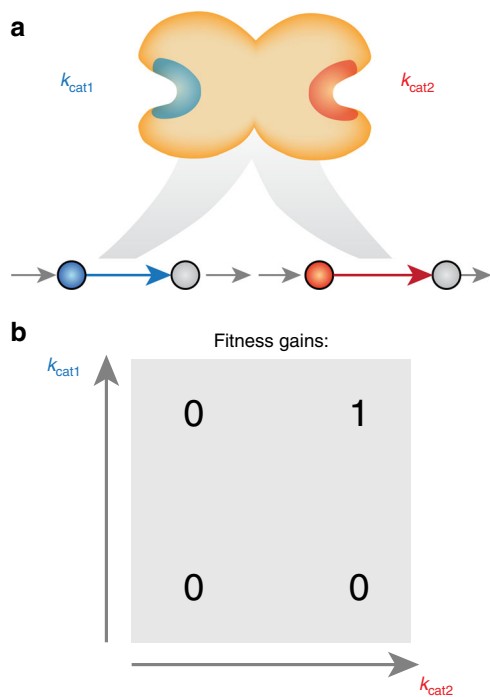

**Fig. 3** Multifunctional enzymes cause synergistic epistasis in $k_{cat}$ evolution. **a** A multifunctional enzyme with two distinct active sites catalyzes two reactions in the same linear fitness-relevant pathway. **b** Mutations that increase either $k_{cat}$ individually cannot be used to reduce protein cost of the pathway and thus exhibit synergistic epistasis

environments to model a more realistic environmental diversity. We randomly chose a set of environmental carbon, nitrogen, and sulfur components, as well as random availability of oxygen (see Methods) and compared prediction performance of this diverse environment simulation with the simulations under constant aerobic glucose conditions.

In vitro measurements of $k_{cat}$ were previously mined from the BRENDA database and filtered for natural substrates[16]. We compared the simulated end points for both constant and diverse environments to this dataset while focusing on reactions without data-driven biophysical constraints to avoid circular conclusions. We found that the predictions agree in magnitude (Fig. 4a, Supplementary Fig. 11 A) and show a significant correlation (Pearson's $R = 0.37$, $p < 6e-4$ for diverse environments. $R = 0.25$, $p < 0.02$ for aerobic growth on glucose. See Methods) with the in vitro data (Fig. 4b, Supplementary Fig. 11B). Simulation of evolution in diverse environments thus results in a better agreement with in vitro data. In addition to in vitro measurements, estimates of in vivo maximal turnover rates ($k_{app,max}$) became recently available based on the combination of proteomics data and flux predictions across multiple conditions[39]. The predicted $k_{cat}$s from both diverse and constant evolutionary environments agree with this in vivo data in magnitude (Fig. 4c, Supplementary Fig. 11C) and show a highly significant correlation ($R = 0.67$, $p < 5e-29$, for diverse environments. $R = 0.57$, $p < 2.4e-19$ for aerobic growth on glucose. See Methods). Like in the case of in vitro measurements, a model of diverse environments explains in vivo data better than constant environments.

What factors affect the speed of evolution of a reaction's $k_{cat}$ until system-wide DRE prevents further adaptation? We find that the $k_{cat}$s in the end points of evolution in diverse environments are correlated with enzyme molecular weight ($R = 0.28$, $p < 4.4e-6$.

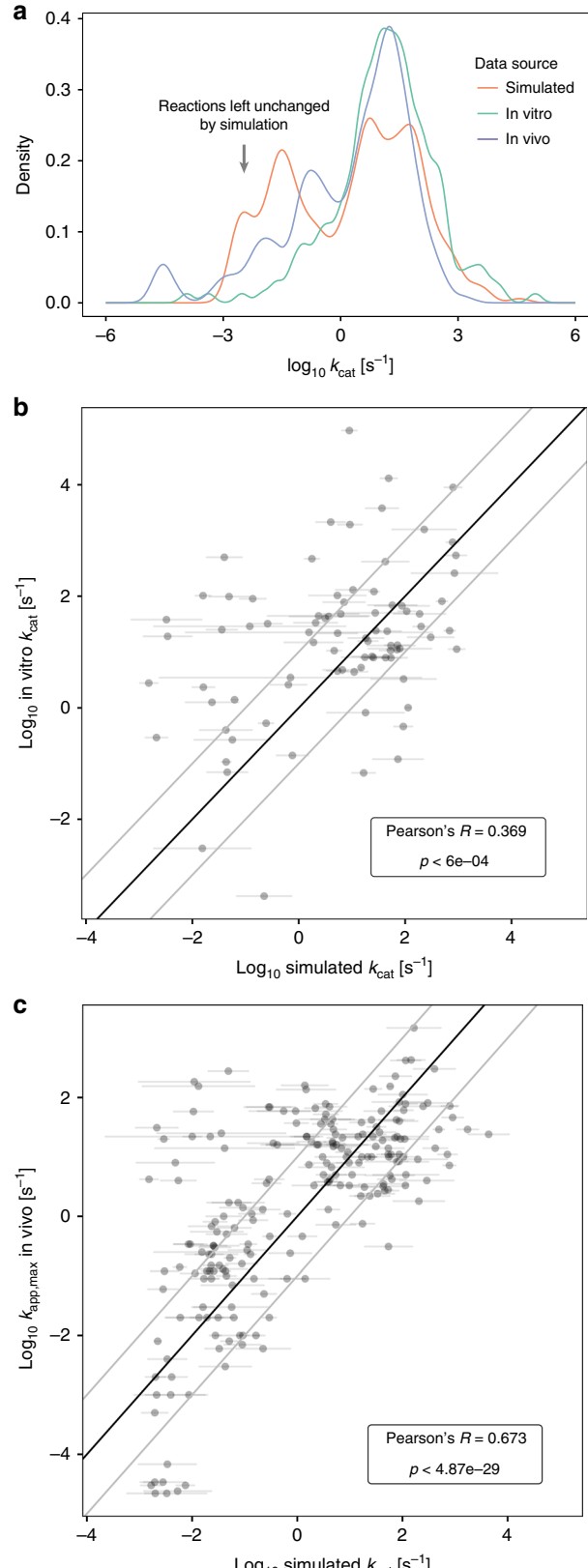

**Fig. 4** Comparison between $k_{cat}$ predictions for evolution in diverse environments and experimental data. **a** Distribution of turnover rates in in vitro ($n = 188$)[16], in vivo ($n = 210$)[39], and simulated data ($n = 276$). Simulated data are only shown for non-constrained reactions that contribute to growth. The arrow indicates reactions that were essentially left unchanged by the simulation, indicating that they were not used in most environments. **b** Comparison between experimental in vitro data and simulated data for reactions contributing to growth ($n = 83$). **c** Comparison between experimental in vivo data ($k_{app,max}$) and simulated data for reactions contributing to growth ($n = 210$). The outliers in the upper left suggest that these reactions are rarely used in the environmental conditions that we model. Horizontal error bars in **b** and **c** show the standard deviation across three simulated replicates. The $p$-values in **b** and **c** are based on Pearson's $R$ to test for significant correlation with a two-sided $t$-test (see Methods). See Supplementary Figure 13 for sensitivity analysis against assumptions about the ancestral state and Supplementary Figure 15 for sensitivity analysis against reaction stoichiometries

in central metabolism are associated with high in vitro $k_{cat}$s[16]. When we repeat our evolutionary simulations in models with random perturbations of reaction stoichiometries and biomass components, agreement with experimental data are abolished (Supplementary Fig. 15). This result confirms the important role of reaction flux as a selection pressure in $k_{cat}$ evolution.

Finally, the convergent behavior we found for evolution in a static environment (Supplementary Fig. 1) is also present in the end points of evolution in diverse environments (all Pearson's $R > 0.87$ across three replicates).

## Discussion

The turnover numbers of enzymes in central energy metabolism are significantly higher than those of pathways in amino acid, fatty acid, nucleotide, and secondary metabolism[16], even though phylogenetic evidence suggest that the core of the metabolic network is conserved across the tree of life[30,31] and extensive enzyme optimization should thus have had sufficient time to occur. In order to understand the mechanistic reason for this observation, we developed an in silico model that predicts the dynamics and long-term end point of $k_{cat}$ evolution, and validated these predictions with experimental data.

It has been suggested that the suboptimal turnover number of many enzymes is the result of an increasing difficulty to achieve $k_{cat}$ improvements that occurs in all metabolic genes[16]. We show that even without such intragenic constraints, a small number of biophysically constrained reactions are sufficient to cause diminishing returns epistasis in otherwise unconstrained reactions (Fig. 2, Supplementary Fig. 7, Supplementary Note 1). As the fitness gain of improvements in $k_{cat}$s (i.e., their selection coefficient $s$) decreases, it approaches the neutral boundary that lies around $1/N_e$[10,11,41], and mutations that yield large improvements in $k_{cat}$ are rendered effectively neutral. Metabolic control theory[42] has been used in the past to postulate the occurrence of diminishing fitness returns when the activity of a single enzyme changes, e.g., explaining the genetic dominance of metabolic genes[43] and the frequency of neutral mutations[41]. In our framework, that situation is comparable to assigning a single reaction to the unconstrained set.

Diminishing returns are often implicitly assumed in quantitative models of adaptation, e.g. in the form of Gaussian fitness landscapes[13], and our results on $k_{cat}$ evolution give a mechanistic example of how diminishing returns can arise, even when the population is still distant from a global optimum. In terms of experimental data, intergenic diminishing returns epistasis has

See Methods) and with the mean of fluxes of parsimonious FBA[40] across diverse growth environments ($R = 0.62$, $p < 2.2e{-}16$. See Methods), indicating that these two factors are the major determinants of selection pressure on a given reaction. This finding explains why the enzymes that catalyze high flux reactions

been found to play a crucial row in a long-term evolutionary experiment[44] and in adaptation to heterologous pathway optimization[45]. In the latter example, the expression cost of a heterologous pathway was reduced by reducing over-expression, a process conceptually similar to the reduction of protein costs through the increase in kinetic parameters. Whereas the adjustment of expression levels is a mechanism commonly found in experimental evolution, kinetic parameter evolution is a smaller mutational target and thus more difficult to study in such a framework.

Structural genomics studies have found convergent evolution of function to be a common pattern in enzyme evolution[46]. Our model shows that kinetic parameter evolution is likely to similarly exhibit convergent behavior. The evolutionary end points show a high correlation of $k_{cat}$s across replicates—even though some reactions diverge—(Supplementary Fig. 1), and final growth rates are very similar (Fig. 2). This suggests a smooth single-peaked phenotypic fitness landscape, where the low level of divergence indicates a plateau of comparable fitness that is reached in a repeatable and convergent manner. Pairwise averaging of end point $k_{cat}$s shows that these intermediate points are also intermediate in fitness (Supplementary Fig. 12), thus confirming the lack of fitness valleys between end points. Remarkably, this high level of convergence is even found when environments differ during the adaptation process (all $R > 0.87$ between end points, also see Fig. 4). As our analysis of end point $k_{cat}$s indicates that selection pressure is mostly determined by flux and—to a lesser extent—enzyme molecular weight, convergence might be caused by correlated flux distributions across environments. We calculate the correlation of flux across 10,000 environments chosen by our sampling algorithm (see Methods) and find a median Pearson correlation of 0.7 between flux distributions on log scale, indicating that this similarity in flux underlies the observed high level of convergence.

Even though diminishing returns epistasis arises for the growth rate effect of mutations, epistatic effects of mutations in the same gene are not modeled explicitly. Thus, even though structural models argue against this[47], intragenic sign epistasis—where the sign of a mutation's effect depends on the genetic background—could cause a more rugged landscape.

Although the model suggests a remarkably smooth fitness landscape, multifunctional enzymes cause "neutral plateaus" that slow adaptation by requiring a neutral mutation to occur before $k_{cat}$ improvements can yield fitness gains (Fig. 3): when removing reactions catalyzed by the product of these genes, fitness jumps are drastically reduced, and the speed of adaptation increases (Supplementary Fig. 6). Most of these cases are caused by multifunctional enzymes that possess two distinct active sites and that have likely resulted from gene fusion events—e.g. *purH*[48] and *histb*[49]. It is thus likely that these gene fusion events occurred after the individual gene products had been selected for higher $k_{cat}$s. Gene fusions are highly polyphyletic[50–52], a finding that supports this idea.

Further genes associated with jump behavior catalyze multiple reactions using the same binding site—e.g., *fabG* (Supplementary Table 1). Kacser and Beeby[33] discussed the effect of such multifunctional enzymes for a scenario of highly un-specific protoenzymes, where gene duplication becomes necessary to render increased specificity adaptive. Nevertheless, the mechanism Kacser and Beeby[33] proposed requires assumptions about how mutations affect each catalytic activity, where experimental data indicate that such effects have to be studied on a case-by-case basis[32]. For the case of multifunctional enzymes that result from gene fusion events, independent mutation effects on both active sites seem a reasonable assumption.

A variety of sources of uncertainty make it difficult to predict experimental $k_{cat}$ data with the ab initio approach we present. Condition-dependent metabolite levels and enzyme affinities (i.e., the $K_m$ values) will affect enzyme saturation where our model assumes full saturation. Undersaturation is thus expected to influence $k_{cat}$ evolution by increasing the selection pressure on $k_{cat}$. A similar effect is expected for the backward flux in thermodynamically unfavorable reactions; e.g., the simulations predict a $k_{cat}$ for the thermodynamically unfavorable malate dehydrogenase reaction of 805 s$^{-1}$ that underestimates in vitro data (931 s$^{-1}$ [53]), whereas in vivo data suggest a much lower effective turnover rate of 7 s$^{-1}$ [39], probably caused by substantial backward flux[39]. Whereas computational feasibility will be a challenge, modeling the interaction between $k_{cat}$, $K_m$, metabolite concentrations, and allosteric regulation is a promising topic for future studies that could also shed light on the co-evolution of isozymes that often vary in $K_m$[54]. As gene duplication is frequently observed in short-term adaptation[55], we assume that most $k_{cat}$s evolved before isozymes emerged and model $k_{cat}$ mutation at the reaction level. Furthermore, our model has to make an assumption about the identity of biophysically constrained reactions. Whereas EC numbers serve as a first approximation for estimating this set, there is still a high level of uncertainty in its true identity. It is in fact possible that a growth-limiting process outside of metabolism causes diminishing returns epistasis, e.g., the expression machinery of the cell. Encouragingly, sensitivity analyses indicate that the qualitative adaptation dynamics and agreement of simulated $k_{cat}$s with experimental data are robust against the identity of the constrained set (Supplementary Figs. 7 and 8, Supplementary Table 2). As studies shed more light on the nature of intragenic fitness landscapes[56], it will be valuable to model the relative contribution of intergenic and intragenic diminishing returns in more detail. The effect of $K_m$ and allosteric effects mentioned above might affect the shape of the inferred fitness landscape; e.g., $k_{cat}$ and $K_m$ frequently show trade-offs[57], a factor that might result in local optima on the fitness landscape. Other sources of uncertainty lie in the choice of selective environments and the shape and parameters of the distribution of mutation effects. Again, sensitivity analyses show that our results are robust against these factors (Supplementary Figs. 9 and 11). As decreases in $k_{cat}$ are expected to be either fitness-neutral or deleterious, they are associated with very low fixation probabilities. Thus, even though we assume mutations that decrease $k_{cat}$s to occur a hundred times more frequently than those that increase $k_{cat}$, only 1.8% of fixed mutations decrease $k_{cat}$s in our evolutionary simulations of varying environments. When ancestral $k_{cat}$ vectors are sampled randomly from the empirical distribution of $k_{cat}$s, the correlation of end points with experimental data decreases ($k_{cat}$ in vitro: $R = 0.29$, $p < 0.007$; $k_{app,max}$: $R = 0.5$, $p < 2e{-}14$; Supplementary Fig. 13, see Methods) as well as the degree of convergence between end points (mean $R^2 = 0.26$, Supplementary Fig. 14, see Methods). This effect is due to the slow accumulation of deleterious mutations that is negligible on the timescale tractable for our simulations—reactions that have a high initial $k_{cat}$ assigned are very unlikely to have substantially decreased it in the end point, even if the reaction is not used in the simulated conditions (Supplementary Fig. 14).

Finally, the strong-selection-weak-mutation regime (SSWM) we use to model adaptation dynamics does not account for the effects of clonal interference, like a decreased rate of adaptation and higher fitness gains of fixed mutations[58]. As the occurrence of diminishing returns are independent of the mutation dynamics, we do not expect clonal interference to have a large effect on end point $k_{cat}$ distributions, although it could prove to be important in future studies quantifying the timescale of $k_{cat}$ fixation.

To validate the assumptions of our modeling approach we compared model predictions to in vitro and in vivo datasets. Despite the sources of uncertainty listed above and the high level of noise in the experimental data (see Bar-Even et al.[16] for discussion) we found a significant agreement with in vitro data and in vivo estimates, where the model explained about 45% of the observed variance in in vivo $k_{cat}$s. In vitro $k_{cat}$s were shown to correlate with enzyme molecular weight and reaction flux ($R = 0.22$ and $R = 0.45$, respectively[4]). Similarly, predicted $k_{cat}$s in our model for diverse environments are correlated with enzyme molecular weight ($R = 0.28$, $p < 4.4e{-}6$) and with the mean of fluxes of parsimonious FBA[40] across diverse growth environments ($R = 0.62$, $p < 2.2e{-}16$). This result indicates that enzyme usage and size determine the selection pressure on individual reactions and thus the magnitude of final $k_{cat}$s, a hypothesis that we confirmed by sensitivity analysis: randomly perturbing network stoichiometry, biomass components, and enzyme molecular weights abolishes the correlation with experimental data (Supplementary Fig. 15). Surprisingly, we found agreement not only by correlation, but also by magnitude (Fig. 4a). This finding is consistent with the realistic growth rates to which the adaptation process converges (Fig. 2). The in vivo data used are based on quantitative proteomics data and flux estimates that assume growth maximization[39]. The better agreement of our simulations with in vivo data might be due to the latter being less noisy than in vitro estimates, but in vivo data could also be biased to prefer our model-based predictions, as model-derived fluxes were used in combination with proteomics data to derive $k_{app,max}$[39]. Nevertheless, using the limited flux data available from metabolic flux analysis (MFA) instead of model-derived flux, a high correlation with model-derived $k_{app,max}$ was found ($R^2 = 0.85$)[39]. Sensitivity analyses (Supplementary Figs. 7 and 9) and our minimal model (Supplementary Note 1) show that the magnitude of evolved $k_{cat}$s can depend on the size of the evolving set, the distribution of mutational effects, and the magnitude of biophysical constraints (Supplementary Fig. 10). We thus provide a consistent set of these parameters, but additional data are required to confirm this parameter set in the future.

In summary, the presented models suggest the following mechanism for $k_{cat}$ evolution: initially, ancestral inefficient enzymes are under strong selection to increase their $k_{cat}$ in order to reduce the protein costs of metabolism. This selection pressure increases with the average flux through the respective reaction and—to a lesser extent—with the molecular weight of the catalyzing enzyme. As soon as some growth-relevant reactions do not have mutations available that could increase their $k_{cat}$—i.e., the reaction becomes biochemically constrained—diminishing returns epistasis affects all other enzymes in the network, and the extent of these diminishing returns is more pronounced in large networks (Supplementary Note 1). Reactions that carry high flux, e.g., those in primary carbon metabolism, still yield substantial fitness benefits and evolve faster than low-flux reactions. Nevertheless, the extent of diminishing returns increases with each mutation that improves a reaction's $k_{cat}$ until selection coefficients become too small to distinguish beneficial from neutral mutations and adaptation comes to a halt. The evolutionary end points exhibit fitness levels that are far lower than theoretically possible states, a property associated with large metabolic networks (Supplementary Note 1).

The prediction of evolutionary outcomes is an ultimate goal in evolutionary biology[9]. The model we present predicts data on $k_{cat}$ in terms of correlation and magnitude, showing that evolutionary long-term end points of $k_{cat}$ evolution can be predicted using evolutionary systems models with considerable accuracy, especially given the sources of model uncertainty listed above. The model predicts that diminishing returns epistasis keeps $k_{cat}$s—and thus fitness—far from the global optimum, indicating the potential of engineering strategies for more efficient enzymes. Whereas we chose E. coli as a model organism to study $k_{cat}$ evolution, the patterns we find are likely to generalize across the tree of life, where organisms with smaller effective population size than E. coli can be expected to show an even stronger mark of insufficient selection in their catalytic properties.

Optimality assumptions are a promising tool for understanding complex biological systems, but finite population sizes and epistatic interactions can render individual molecules far from theoretical optima—even when the underlying fitness landscape is smooth. Seeing cells through the systems perspective and modeling evolutionary history can be crucial for understanding cell behavior, as is the case for kinetic turnover numbers.

## Methods

**Growth rate predictions using MOMENT**. In the simulation of kinetic parameter evolution, the growth rate that results from a given vector of catalytic turnover rates κ is predicted using the MOMENT algorithm[4]. MOMENT is conceptually similar to flux balance analysis (FBA[59]), in that it maximizes the growth rate $\mu$ by maximizing flux into a biomass reaction ($v_z$) given a set of constraints ($\mathbf{v}_{min}$ and $\mathbf{v}_{max}$):

$$\max(v_z)\, s.t.$$

$$\mathbf{Sv} = 0$$

$$v_{min,i} \leq v_i \leq v_{max,i}.$$

Here, $\mathbf{S}$ represents the stoichiometric matrix and $\mathbf{v}$ the vector of fluxes. MOMENT extends FBA by introducing enzyme concentrations as model variables ($g_i$, mmol $g_{DW}^{-1}$) and recursively parsing gene-protein-reaction (GPR) rules to obtain upper limit constraints on metabolic fluxes:

$$v_i \leq f(\kappa_i, G_i),$$

where $G_i$ represents the set of genes involved in catalyzing reaction $i$. The respective GPR is parsed by using the maximum of enzyme concentrations to represent AND relations and the sum to model OR relations. Finally, the total weight of the metabolic proteome ($C$, $g_{protein}\, g_{DW}^{-1}$) and the respective enzyme molecular weights (MW) are used to constrain enzyme concentrations:

$$\sum g_i \mathrm{MW}_i \leq C.$$

MOMENT was used to simulate growth in iJO1366, a genome-scale model of E. coli K-12 MG1655 metabolism[25]. Enzyme molecular weights were calculated based on the E. coli K12 MG1655 protein sequences (NCBI Reference Sequence NC_000913.3), and $C$ was set to 0.32 $g_{protein}\, g_{DW}^{-1}$ in accordance with the E. coli metabolic protein fraction across diverse growth conditions[4,60]. Linear programming problems were constructed using the R[61] packages sybil[62] and sybilccFBA and solved using IBM CPLEX version 12.7. The growth rate $\mu$ (compare Fig. 1) can then be obtained as the flux into the biomass reaction $v_z$.

We classify a reaction as contributing to in silico growth using flux variability analysis[38]. When either the maximal flux or the absolute minimal flux through a reaction that still optimizes the growth rate $\mu$ in FBA is $>10^{-6}$ mmol $g_{DW}^{-1}\, h^{-1}$, we call a reaction "contributing to growth in silico".

**An MCMC algorithm for simulating $k_{cat}$ evolution**. We assume a genetically homogenous population of cells with a population size equal to the effective population size estimated for E. coli ($N_e = 2.5e7$[35]). A single iteration of the Markov Chain Monte Carlo (MCMC) algorithm starts as follows: A mutation affecting the $k_{cat}$ of a single randomly chosen reaction $i$ is simulated as multiplying an original $k_{cat}$ ($= \kappa_i$) by a factor α that is drawn from a lognormal distribution with mean and standard deviation in log scale $\log(3/2)$ and 0.3, respectively. This distribution determines the jump size in the space of $k_{cat}$s, but not the ratio between deleterious to advantageous mutations (see below).

$$\kappa_{i,mut} = \alpha \kappa_i.$$

As formulated by the Haldane relationship[63], $k_{cat}$s of forward and backward directions and respective $K_m$s cannot change independently from each other. To account for the Haldane relationship, we implement mutations that affect the forward and backward $k_{cat}$ of reversible reactions equally. The growth rate of the original strain ($\mu$) and the strain carrying the mutation affecting $\kappa_i$ ($\mu_{mut}$) is then

calculated by solving the MOMENT problem detailed above (also see Fig. 1). Assuming that fitness is proportional to growth rate, we can obtain the selection coefficient $s$ and the fixation probability $\pi$[36]:

$$s = 1 - \frac{\mu}{\mu_{\text{mut}}},$$

$$\pi = \begin{cases} \frac{1}{N}, & \text{if } s = 0 \\ \frac{1 - e^{-2s}}{1 - e^{-2Ns}}, & \text{otherwise} \end{cases}.$$

The fixation probability $\pi$ is then used to decide the fixation of the novel mutation. In case of a successful fixation event, the vector of $k_{\text{cat}}$s, $\kappa$, is updated at position $i$ with the newly fixed mutation, or, in case of an unsuccessful fixation event, the previous $\kappa_i$ remains the most abundant allele. The next iteration of the algorithm starts with introducing a novel change in the $k_{\text{cat}}$ of a random enzyme, and so on. A typical simulation run simulates around $10^8$ mutations that have the chance to become fixed, requiring $10^8$ linear programs to be solved for a single replicate.

The high population size allowed us to optimize simulation performance by heuristically setting the ratio of deleterious to advantageous mutations: the growth rate for a deleterious mutation was simulated once, but their fixation was sampled multiple times to arrive at a 100:1 ratio between deleterious and advantageous mutations (see Supplementary Table 3 for sensitivity analysis). Certain reaction mechanisms were shown to consistently exhibit low $k_{\text{cat}}$s[16]. We use the enzyme commission (EC) number to set the reactions belonging to the three (out of six) top level codes with the highest median in vitro $k_{\text{cat}}$—namely oxidoreductases, hydrolases, and isomerases—as biophysically unconstrained. In order to allow an unbiased comparison with experimental data, all reactions for which data was available were also set as unconstrained. The remaining reactions were considered biophysically constrained and were fixed to the median of in vitro $k_{\text{cat}}$ measurements (13.7 s$^{-1}$). The $k_{\text{cat}}$s of unconstrained reactions were initialized to $10^{-3}$ s$^{-1}$. See Supplementary Figures 7 and 8, and Supplementary Table 2 for sensitivity analysis against the identity of the constrained set.

In order to simulate diverse environments, we applied random sampling of a new environment every 1000 iterations. Here, oxygen uptake was allowed with probability 1/2, and the environment always contained at least one randomly chosen source of each carbon, nitrogen, sulfur, and phosphate. A number of additional sources were drawn from a binomial of size 2 with success probability 1/2. This process was repeated until a growth sustaining environment was found and the following 1000 mutations were simulated in this novel environment.

**Statistics**. Pearson's $R$ was used to test for significant correlation with a two-sided $t$-test as implemented in the cor.test() function of the R environment[61].

**Code availability**. R code for the simulations presented in this study is available from the authors upon request.

## Data availability
Predicted $k_{\text{cat}}$ end points that are presented in this study are available from the authors upon request.

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

## Acknowledgements

The authors would like to thank Abdelmoneim Amer Desouki for his support in using the sybilccFBA package, and Ron Milo and Laurence Yang for helpful discussion. This research used resources of the National Energy Research Scientific Computing Center, a DOE Office of Science User Facility supported by the Office of Science of the U.S. Department of Energy grant number DE-SC0008701. This work was supported by the Novo Nordisk Foundation grant number NNF10CC1016517.

## Author contributions

D.H., D.C.Z., and B.O.P. designed the study. D.H. conducted all modeling, simulation, and data analysis. D.H., D.C.Z., and B.O.P. wrote the paper.

## Additional information

**Competing interests:** The authors declare no competing interests.

