## [Peer Review File · Nature Communications]

Reviewers' comments:

Reviewer #1 (Remarks to the Author):

I hereby submit my review report on the manuscript: "Modeling genome-wide evolution of catalytic turnover rates: Strong epistasis shaped modern enzyme kinetics" by David Heckmann and coworkers. In the paper, the authors take a modeling based approach to try and explain why the turnover numbers, k_{cat} , of many enzymes evolved to be considerably lower than their theoretically predicted maximums. The authors suggest that increasing catalytic turnover numbers is subject to diminishing returns in terms of the acquired fitness. This means that beyond a certain point mutations that result in a very large increase of k_{cat} will only confer a tiny gain of fitness on the organism as a whole. These mutations will not be able to fix in a finite population and will eventually be washed away by genetic drift.

To demonstrate that this mechanism could have potentially played a role in shaping turnover numbers as we know them the authors performed in-silico evolution experiments. They showed that constraining the evolution of a subset of enzymes in a biochemical network subjects the evolution of the complementary, and otherwise unconstrained, set to diminishing returns. These result in sub-optimal k_{cat} s, and the authors are moreover able to demonstrate substantial correlation between terminal k_{cat} values achieved in silico and those known to us from the literature.

The work of Heckmann and coworkers is an interesting and original development that is of interest to the scientific community, and I moreover believe that it could give rise to several follow up investigations. I am thus positive towards this paper, but also feels that it requires additional work to be done before I will be able to recommend it for publication in Nature Communications. My concerns are specified below.

1. Right at the abstract, the authors declare that: "The resulting fitness landscape is smooth and causes k_{cat} evolution to be convergent." They then repeat this claim at various places throughout the paper, e.g. in lines 243-247 where they state: "The evolutionary end points show a high correlation of k_{cat} s across replicates—even though some reactions diverge—(Supplemental Figure 1), and final growth rates are very similar (Figure 2). This suggests a smooth single-peaked phenotypic fitness landscape, where the low level of divergence indicates a plateau of comparable fitness that is reached in a repeatable and convergent manner."

The shape of the phenotypic fitness landscape investigated by the authors is of considerable interest to the community. The authors provide some evidence for its flatness in the vicinity of the evolutionary end points (terminal k_{cat} values), but I would like them to further investigate this issue to decide if evolutionary end points really sit together on a plateau or rather on two or more isolated high regions that are separated by fitness valleys. To determine this, I propose the following. For every pair of evolutionary end points in the multidimensional k_{cat} space find the corresponding midpoint and determine its fitness. In other words, determine the fitness of the entire set of hybrid organisms that are generated by averaging two k_{cat} vectors (parent organisms) that emerged from the original simulation. If all hybrid organisms will turn out to have

similar fitness values to those of their parents, one could conclude that the terminal fitness landscape is indeed flat. On the other hand, if it will be found that some of the hybrid organism have fitness values that are substantially lower than those of their "parents", this would indicate that the two "parent" end points are actually separated in fitness space. One could then cluster the data to try and assess the number of distinct high regions on top of which terminal end points reside.

Let me stress, that while the authors originally suggested that the terminal fitness landscape is flat – publication of their paper should not be prevented even if the analysis suggested above would reveal otherwise. I am saying this to encourage the authors to try and really get down to the bottom of this important question with the aim of providing a less ambiguous picture of the underlying fitness landscape.

2. In Figure 2A, the authors provide a few evolutionally trajectories to illustrate their progression with time. Presenting raw simulation data is important, but a more thorough analysis of this data is missing here. For example, I would like the authors to plot the average evolutionally trajectory to determine how the terminal growth rate is approached. This could help decide, for example, if convergence is exponential in time (number of mutation events) or slower from some reason. I would also like to see a more detailed analysis of variability round and about the mean trend line. Replicates start with similar identical growth rates and evolve to attain similar terminal growth rates, but variability in growth rate is clearly discernible at intermediate time points. How does this variability evolve with time? And what could we possibly learn from this? Analysis and discussion of these important questions is currently missing from the manuscript.

3. In line 170, the authors state that "The high level of convergence that is exhibited in the adapted growth rates (Figure 2A) is reflected in the turnover rates of the evolved populations...". However, at this point it is not clear if this result truly reflects an interesting property of the fitness landscape or if it simply mirrors the fact that initial conditions for all populations in the sample were identical. Adding a small amount of noise to the initial conditions would probably not change the result too much, but I would like to know what happens if initial conditions for the k_{cat} s are randomly sampled from a distribution that closely mimics the distribution observed in vivo and in vitro (reference 15 & 37 in the paper). Note that this does not mean that specific k_{cat} values are chosen to mimic the observed ones, but rather that the statistical distribution of k_{cat} values is similar. Would evolution show the same level of convergence in this case as well? If not, why?

4. In line 222, the authors state that: "...a small number of biophysically constrained reactions are sufficient to cause diminishing returns epistasis in otherwise unconstrained reactions (Figure 2, Supplemental Figure 5, Supplemental Information 2)." However, from my understanding, this study was performed under conditions where the number of biophysically constrained reactions was actually comparable to the number of unconstrained reactions. Regardless, I would like to see the authors investigate this issue further to quantify how terminal turnover rates depend on the fraction of biophysically constrained reactions. The authors have already shown that when these are completely lacking growth rates can grow indefinitely, and that terminal growth rates will be limited when the fraction of constrained reactions is substantial. A more quantitative analysis of the transition between these two extremes is, however, lacking and should be added to the paper.

5. In line 290, after pointing out substantial correlation between the k_{cat} values evolved in silico and those taken from various places in the literature, the authors indicate that: "As our model uses information on reaction mechanisms only for the identification of the biophysically constrained set, this agreement confirms the hypothesis that these correlations are induced by differential selection pressures". However, in order to truly convince that this is indeed the case, the authors should compare the correlations obtained to a null-hypothesis scenario in which little to no correlations are expected to arise. In particular, the author should check to see what happens when the biophysically constrained set is not chosen based on prior knowledge, but rather drawn at random from the entire pool of available reactions. Doing this is the only way to be certain that the real set of biophysically constrained reactions holds in it relevant information that is crucial to the determination of terminal k_{cat} values in the unconstrained set.

6. Finally, one could not help but wonder about the place of the Michaelis constant K_m . This is

entirely lacking from the paper as the authors explicitly state that: "Condition-dependent metabolite levels and enzyme affinities will affect enzyme saturation where our model assumes full saturation...". However, could the authors perhaps offer a way in which K_m be integrated into similar future studies? Perhaps, with the aid of an appropriate model, the value of this important parameter could also be predicted?

Minor concerns and typos

1. In line 43 the authors write: "...experimental data on k_{cat} is scarce...". In what way is it scarce? The authors should clarify this statement in light of the tens of thousands of kinetic parameters that could currently be found at the Enzyme Database – BRENDA.
2. In line 130, $1.58s^{-1}$ should probably be $1.58h^{-1}$.

Reviewer #2 (Remarks to the Author):

The paper presents a model to understand the evolution of enzyme turnover numbers. The model combines the genome-scale models of metabolic networks with population genetics models and provides a systems level view of k_{cat} evolution. The paper is very well written and I think the authors have covered all the aspects required for the analysis and have stated their assumptions clearly. Few minor comments.

1. The authors discuss the scenario when an enzyme is catalysing multiple reactions (multifunctional enzymes). However, in many cases one reaction is catalysed by multiple enzymes. It would be useful to see how their model would behave in such cases.
2. On Line: 154 the authors state that "small set of genes that was repeatedly responsible for large jumps in fitness". For this, they set all enzymes unconstrained resulting in the increase in growth rate. I was wondering if they tried simulating the model while keeping these enzymes/genes in the constrained set and compare the trajectories with the reported model.
3. Line 267: It wasn't clear to me if the authors have included the effects of inhibitors and enhancers on the enzymes. To the best of my understanding, they haven't. If so, the authors might want to mention that they are ignoring these effects.

Typos/minor errors

=====

1. Line # 119: Figure 2C does not exist in the manuscript
2. Figure SI2 1: different terminologies used for the same reaction (biophysically and biochemically).
3. Line # 157: "(purH), and cell ..." should be "(purH), cell"
4. Line # 188: data set should be dataset
5. Line # 635: of the number of the number of (repetition)

Reviewer #3 (Remarks to the Author):

This paper presents an interesting and innovative way of investigating the evolution of metabolism. The authors show how the effect of mutations that affect the catalytic rate of enzymes depends on the genetic background of the organism; their model enables them to identify changes in k_{cat} that have a strong beneficial effect (multifunctional and enzymes with a

low current k_{cat}). It also becomes clear how a small number of biophysical constraints can determine the maximally achievable growth rate of a cell.

Although the approach is interesting, some of the model assumptions can be questioned and in view of the assumptions, some of the model results are fairly trivial and would not need the brute-force approach of a Genome-Scale Model.

Model assumptions

All enzymes are fully saturated. The authors cite Bennet et al. (2009) to argue that this assumption is reasonable. It should however be noted that cells were grown on saturating amounts of nutrients in this study and that Bennet et al. did find that the enzymes in lower glycolysis were not saturated. Making this assumption and thereby discarding effects like product inhibition and allosteric effects, strongly simplifies the dependence of the growth rate on k_{cats} which is likely the reason that the authors find a smooth fitness landscape with only one maximum. What would happen if inequalities were to be used (i.e. $v \leq k_{cat} \cdot e$), allowing for sub-saturation?

Additionally, substrate specificity and turnover rates are constrained by thermodynamics of the catalyzed reaction via microscopic reversibility - the Haldane relationship. By assuming full saturation of all enzymes, this constraint is relieved, obviously, but there are quite a number of papers –e.g. from Ron Milo's lab- showing the importance of thermodynamics in defining the costs associated with metabolic flux.

The identity of biophysically constrained reactions is assumed known. I agree with the authors that their modelling framework is a nice starting point and that choosing the set of biophysically constrained reactions based on EC numbers is a good first approximation. However, we can't be sure that enzymes that have low current k_{cats} are slow because of biophysical constraints or because of the very evolutionary history that the authors try to model. Using this set of constrained reactions therefore imposes part of the answer on the model. The authors also state that their results are robust against the identity of the constrained set. This is, in my opinion, only partly true:

In Supplementary Information 2 (e.g. Equation on line 636) the authors show that in a linear pathway the eventual growth rate is largely determined by the number of constrained reactions. Since it was mathematically proven (Wortel et al., Müller et al. (2014)) that resource allocation models such as MOMENT attain maximal growth rate in an Elementary Flux Mode (for which the authors' derivation is equally valid as for a linear pathway), this result is true even in the genome-scale modelling case. It is therefore almost trivially true that the achieved growth rate is robust against the identity of the set, but not against the size of the set.

The actual important validation of the results - for which robustness should thus be tested- is the correlation of simulated k_{cats} and experimental k_{cats} . I suspect that this correlation is not robust at all against the identity of the set of constrained reactions.

Results

A smooth single-peaked phenotypic fitness landscape is found. This is a direct consequence of the model setup. The growth rate is a monotone function of all k_{cats} . At the same time, there is a fundamental upper bound to the growth rate, derived in SI 2. Given enough iterations, all unbounded k_{cats} will become as large as possible (although the rate of increase will slow down enormously). The Genome Scale Model is not needed for this result, and in fact masks the triviality of the outcome.

The model predictions do correlate significantly with in vitro k_{cats} . What I miss is an assessment of the robustness or triviality of this result (see also comment above). Would changes in the network stoichiometry, or even random networks, produce similar correlates? The correlation with in vitro data is (just) significant, but not necessarily impressive.

The model predictions are in better agreement with in vivo data than with in vitro data.

The cause of this result is not entirely clear to me, and is also not extensively discussed or explained. To what extent can this be the result of the in vivo data being generated using an FBA-based approach too?

“The evolutionary long-term end points of k_{cat} evolution can be predicted using systems models with unexpected accuracy”. I don’t think such a strong statement is validated when a non-negligible set of k_{cats} is imposed on the model, especially because fixing these parameters strongly influences the predicted endpoints.

Minor typo’s

- The growth rate on line 130 should be per hour, not per second

Reviewer #4 (Remarks to the Author):

This paper outlines an excellent, and very timely approach to simulating metabolic evolution. While the findings are entirely consistent with MCA theory, nobody has ever applied such a broad, grand approach to networks. A tour de force, a long string of excellent points were made, and the writing is quite good, too.

My only real “objection” is with the anti-MCA tone that comes across, as by my reading there is broad similarity in approaches. Consider, for example, the idea that changes in a single enzyme will exhibit diminishing returns. Although you are changing many enzymes, the intuition is essentially the same: the more you fix the fixable problems, the less they are a problem (i.e., their control coefficients go down). Your approach is phenomenally more sophisticated in breadth, and doesn’t rely upon linearization around local changes to the steady-state – so it is distinctly more realistic – but the root cause is not so different. And you should occasionally have seen synergistic (positive) epistasis for single-function enzymes if you have two in a row that hit the same pathway. The first such mutation in enzyme A will decrease its control coefficient but raise that for enzyme B...

Was it really necessary to assume only 1% of mutations increased k_{cat} if you are modeling adaptation under SSWM? Did you ever find BENEFICIAL drops in activity, for example? I know this is possible due to ultrasensitivity around branchpoints, but I think it deserves comment.

The jumps in fitness are quite fun, but I feel that there is a second reason (besides the VERY well-explained constraint brought up) that this can occur in a model but not in an experiment: SSWM versus clonal interference. If you actually had a population making mutations that mimicked the N_e of a flask, and fixations were not instantaneous, then clonal interference between lineages should cause the greatest gains to occur early on rather than later. It would be worth mentioning this aspect, I think. (And maybe it becomes a future paper comparing the difference.)

At first read in the abstract on predicting experimental k_{cat} seemed to make a big claim (that I was quite dubious about). When I got down to that section, I understood the extent of the claim, but it would be useful from the start to say that X portion of the variance was explained. What still isn’t spelled out is why the hell it should work at all! My guess is that the predicted flux values needed for optimal growth create the primary metabolism/secondary metabolism sort of scenario that leads to the need for higher k_{cat} in high flux reactions. Is this indeed the case? Just a bit more explanation about this would be quite helpful, or perhaps show that a distinct assumption – like growth on something like acetate – would lead to a poorer correlation because it would underestimate glycolysis, etc.

Minor comment:

p.5 – Out of curiosity, as it was hilariously ridiculous, what constrains the “unconstrained” case to a doubling time of 1.58 per second? Is that just some computational thing?

Reviewer comments with detailed responses and actions taken

Reviewer #1 (Remarks to the Author):

I hereby submit my review report on the manuscript: “Modeling genome-wide evolution of catalytic turnover rates: Strong epistasis shaped modern enzyme kinetics” by David Heckmann and coworkers. In the paper, the authors take a modeling based approach to try and explain why the turnover numbers, k_{cat} , of many enzymes evolved to be considerably lower than their theoretically predicted maximums. The authors suggest that increasing catalytic turnover numbers is subject to diminishing returns in terms of the acquired fitness. This means that beyond a certain point mutations that result in a very large increase of k_{cat} will only confer a tiny gain of fitness on the organism as a whole. These mutations will not be able to fix in a finite population and will eventually be washed away by genetic drift.

To demonstrate that this mechanism could have potentially played a role in shaping turnover numbers as we know them the authors performed in-silico evolution experiments. They showed that constraining the evolution of a subset of enzymes in a biochemical network subjects the evolution of the complementary, and otherwise unconstrained, set to diminishing returns. These result in sub-optimal k_{cat} s, and the authors are moreover able to demonstrate substantial correlation between terminal k_{cat} values achieved in silico and those known to us from the literature.

The work of Heckmann and coworkers is an interesting and original development that is of interest to the scientific community, and I moreover believe that it could give rise to several follow up investigations. I am thus positive towards this paper, but also feels that it requires additional work to be done before I will be able to recommend it for publication in Nature Communications. My concerns are specified below.

1. Right at the abstract, the authors declare that: “The resulting fitness landscape is smooth and causes k_{cat} evolution to be convergent.” They then repeat this claim at various places throughout the paper, e.g. in lines 243-247 where they state: “The evolutionary end points show a high correlation of k_{cat} s across replicates—even though some reactions diverge—(Supplemental Figure 1), and final growth rates are very similar (Figure 2). This suggests a smooth single-peaked phenotypic fitness landscape, where the low level of divergence indicates a plateau of comparable fitness that is reached in a repeatable and

convergent manner.”

The shape of the phenotypic fitness landscape investigated by the authors is of considerable interest to the community. The authors provide some evidence for its flatness in the vicinity of the evolutionary end points (terminal k_{cat} values), but I would like them to further investigate this issue to decide if evolutionary end points really sit together on a plateau or rather on two or more isolated high regions that are separated by fitness valleys. To determine this, I propose the following. For every pair of evolutionary end points in the multidimensional k_{cat} space find the corresponding midpoint and determine its fitness. In other words, determine the fitness of the entire set of hybrid organisms that are generated by averaging two k_{cat} vectors (parent organisms) that emerged from the original simulation. If all hybrid organisms will turn out to have similar fitness values to those of their parents, one could conclude that the terminal fitness landscape is indeed flat. On

the other hand, if it will be found that some of the hybrid organism have fitness values that are substantially lower than those of their “parents”, this would indicate that the two “parent” end points are actually separated in fitness space. One could then cluster the data to try and assess the number of distinct high regions on top of which terminal end points reside.

Let me stress, that while the authors originally suggested that the terminal fitness landscape is flat – publication of their paper should not be prevented even if the analysis suggested above would reveal otherwise. I am saying this to encourage the authors to try and really get down to the bottom of this important question with the aim of providing a less ambiguous picture of the underlying fitness landscape.

***Response:** We would like to thank the Reviewer for the thoughtful and practical suggestion. We agree that this “hybrid” experiment is an efficient way to better understand the structure of the underlying fitness landscape in proximity of evolutionary endpoints. We conducted the proposed in silico experiment and present the results in Supplementary Figure 11. The analysis confirms that evolutionary endpoints indeed share a flat fitness plateau, with hybrid species outperforming their worst parent in all cases. We discuss these results on page 13 lines 257-258.*

***Actions:** Added Supplementary Figure 11 and discussed results on page 13 lines 257-258.*

2. In Figure 2A, the authors provide a few evolutionally trajectories to illustrate their progression with time. Presenting raw simulation data is important, but a more thorough analysis of this data is missing here. For example, I would like the authors to plot the average evolutionally trajectory to determine how the terminal growth rate is approached. This could help decide, for example, if convergence is exponential in time (number of mutation events) or slower from some reason. I would also like to see a more detailed analysis of variability round and about the mean trend line. Replicates start with similar identical growth rates and evolve to attain similar terminal growth rates, but variability in growth rate is clearly discernible at intermediate time points. How does this variability evolve with time? And what could we possibly learn from this? Analysis and discussion of these important questions is currently missing from the manuscript.

***Response:** We agree that this analysis of average adaptation dynamics is important for understanding the general properties of k_{cat} evolution. We now computed the average evolutionary trajectory and its variability. Variability in growth rates is highest in the intermediate phases of adaptation, where large jumps dominate dynamics. Interestingly, we find that the sigmoidal shape of the average trajectory is*

close to the behavior of the simple single-pathway model that we present in Supplementary Information 2. This shows that, while individual trajectories exhibit irregular adaptation dynamics that are caused by multifunctional enzymes, the average trajectory exhibits the same saturation behavior that is predicted by our simple analytical model. We then fitted the analytical model to the growth trajectories and show a comparison of simulation data, average trajectory, and model fit in Supplemental Figure 3.

Actions: *We added the average evolutionary trajectory and its variability to Figure 2A. We further added Supplementary Figure 3 to show the fitted model in comparison with the average trajectory and simulated trajectories. The results are now described on page 5 line 115-117 of the Results section.*

3. In line 170, the authors state that “The high level of convergence that is exhibited in the adapted growth rates (Figure 2A) is reflected in the turnover rates of the evolved populations...”. However, at this point it is not clear if this result truly reflects an interesting property of the fitness landscape or if it simply mirrors the fact that initial conditions for all populations in the sample were identical. Adding a small amount of noise to the initial conditions would probably not change the result too much, but I would like to know what happens if initial conditions for the k_{cat} s are randomly sampled from a distribution that closely mimics the distribution observed in vivo and in vitro (reference 15 & 37 in the paper). Note that this does not mean that specific k_{cat} values are chosen to mimic the observed ones, but rather that the statistical distribution of k_{cat} values is similar. Would evolution show the same level of convergence in this case as well? If not, why?

Response: *We conducted the proposed in silico experiment and initialize starting k_{cat} vectors to random samples from a distribution that mirrors that of in vitro k_{cat} s. We find that the correlation between end point k_{cat} vectors and experimental data decreases but remains statistically significant. The correlation among end points likewise decreases. The reason for this behavior lies in the low fixation probability of mutations that decrease k_{cat} s. Such mutations are slightly deleterious, and thus their fixation probability is below $1/N_e=2.5e-7$. Thus, reactions that have an unrealistically high initial k_{cat} assigned are very unlikely to have substantially decreased it in the endpoint, even if the reaction is not used in the simulated conditions.*

Actions: *We added Supplemental Figure 12 and 13 that show the effect of random starting vectors on correlation with experimental data and end point convergence, respectively. We discuss these results on page 15 lines 307-317.*

4. In line 222, the authors state that: “...a small number of biophysically constrained reactions are sufficient to cause diminishing returns epistasis in otherwise unconstrained reactions (Figure 2, Supplemental Figure 5, Supplemental Information 2).” However, from my understanding, this study was performed under conditions where the number of biophysically constrained reactions was actually comparable to the number of unconstrained reactions. Regardless, I would like to see the authors investigate this issue further to quantify how terminal turnover rates depend on the fraction of biophysically constrained reactions. The authors have already shown that when these are completely lacking growth rates can grow indefinitely, and that terminal growth rates will be limited when the fraction of constrained reactions is substantial. A more quantitative analysis of the transition between these two extremes is, however, lacking and should be added to the paper.

Response: We agree that the size and identity of the constrained set are crucial and uncertain parameters in our analysis. They affect the speed of adaptation while having little effect on the growth rate after $5e7$ iterations (Supplementary Figure 5 of the original manuscript). We now extend the original results with a detailed analysis of the effect of identity and size of the constrained set on the endpoint k_{cat} s and their correlation with experimental data (Supplemental Table 2 and Supplemental Figure 8). We find that the magnitude of evolved k_{cat} s and their agreement with experimental data are surprisingly robust against the size and identity of the evolving set. Furthermore we conducted a statistical analysis of the effect of set size on the final growth rates, finding no significant differences. We added this result to Supplemental Figure 7.

Actions: We added Supplemental Table 2 and Supplemental Figure 8, and referenced them on page 14 lines 298-299. We further added statistical analysis to Supplemental Figure 7.

5. In line 290, after pointing out substantial correlation between the k_{cat} values evolved in silico and those taken from various places in the literature, the authors indicate that: “As our model uses information on reaction mechanisms only for the identification of the biophysically constrained set, this agreement confirms the hypothesis that these correlations are induced by differential selection pressures”. However, in order to truly convince that this is indeed the case, the authors should compare the correlations obtained to a null-hypothesis scenario in which little to no correlations are expected to arise. In particular, the author should check to see what happens when the biophysically constrained set is not chosen based on prior knowledge, but rather drawn at random from the entire pool of available reactions. Doing this is the only way to be certain that the real set of biophysically constrained reactions holds in it relevant information that is crucial to the determination of terminal k_{cat} values in the unconstrained set.

Response: We agree that the identity of the constrained set is of high interest. To avoid misunderstandings: we do not think the experimental data allows us to identify the true constrained set at this point. Instead, we now show in Supplemental Table 2, Supplemental Figure 7, and 8 that the identity of the random set has little influence on the final state of evolving k_{cat} s; correlation with experimental data and final magnitude are robust. Of course, in the case of a constrained set becoming small enough to not be limiting growth, diminishing returns disappear and growth rates explode, as shown in Supplemental Figure 4. To show that instead the network structure is the main component that allows us to explain experimental data, we randomly perturbed network stoichiometry and biomass components and find that these perturbation indeed abolish the ability of the model to explain the data (Supplemental Figure 14).

Actions: We rephrased our explanation of differential selection pressures (page 15 lines 331-335), referencing the new Supplemental Figure 14. We further added Supplemental Table 3, Supplemental Figure 8 and referenced them in the discussion and added the statistical analysis of final growth rates to Supplemental Figure 7.

6. Finally, one could not help but wonder about the place of the Michaelis constant K_m . This is entirely lacking from the paper as the authors explicitly state that: “Condition-dependent metabolite levels and enzyme affinities will affect enzyme saturation where our model assumes full saturation...”. However,

could the authors perhaps offer a way in which K_m be integrated into similar future studies? Perhaps, with the aid of an appropriate model, the value of this important parameter could also be predicted?

Response: *We agree that undersaturation is likely to have influenced k_{cat} evolution. We omitted the the interaction of metabolite concentrations and K_m from our simulations for three main reasons: (1) Computational feasibility. Every iteration of the presented MCMC algorithm involves the solution of a linear program, representing the majority of computation time. Final growth rates are typically only approached after ~1000 CPU hours for a single replicate. Inclusion of nonlinear Michaelis-Menten kinetics is thus likely to make simulations infeasible for the presented MCMC algorithm. (2) k_{cat} and K_m are likely to have co-evolved (e.g., Savir et al. 2010, PNAS 107, 3475-3480), thus it is warranted to model K_m as an evolving parameter in the model. This would increase the number of unknown model parameters tremendously. If we instead assume that k_{cat} s evolved in enzymes that exhibited modern experimental K_m s, we would induce additional experimental noise on the predicted k_{cat} s. (3) Undersaturation increases enzyme costs and is thus likely to be selected against, suggesting that our assumption introduces little bias in our predictions. Given these reasons, we decided to not include undersaturation and instead focus on the variance in experimental data that can be explained when assuming full saturation.*

Actions: *We now discuss the effect of undersaturation and thermodynamic constraints on k_{cat} evolution in more detail, and explain technical difficulties that future studies will have to face on page 13 lines 282-294.*

Minor concerns and typos

1. In line 43 the authors write: "...experimental data on k_{cat} is scarce...". In what way is it scarce? The authors should clarify this statement in light of the tens of thousands of kinetic parameters that could currently be found at the Enzyme Database – BRENDA.

Actions: *We now mention the model coverage of BRENDA k_{cat} s in E.coli in the introduction on page 2 lines 44-45.*

2. In line 130, 1.58s⁻¹ should probably be 1.58h⁻¹.

Actions: *Corrected this typing error.*

Reviewer #2 (Remarks to the Author):

The paper presents a model to understand the evolution of enzyme turnover numbers. The model combines the genome-scale models of metabolic networks with population genetics models and provides a systems level view of k_{cat} evolution. The paper is very well written and I think the authors have covered all the aspects required for the analysis and have stated their assumptions clearly. Few minor comments.

1. The authors discuss the scenario when an enzyme is catalysing multiple reactions (multifunctional enzymes). However, in many cases one reaction is catalysed by multiple enzymes. It would be useful to see how their model would behave in such cases.

Response: *Our model indeed ignores the effect of isozymes and models k_{cat} evolution at the reaction level rather than the gene level. We intentionally implemented this reaction-specific algorithm for two main reasons: (1) Gene duplication is frequently observed as a mechanism of short-term adaptation (e.g., Romero & Palacio 1997, Annu Rev Genet 31, 91-111) and modern isoforms are thus expected to have formed after kinetics were optimized. (2) Experimental data that would allow isoform comparison is not available: in vitro data is typically not associated with gene-information and $k_{app,max}$ is only available for reactions that are catalyzed by unique homomeric enzymes. That being said, isoenzymes frequently vary in their K_m parameters. It would thus be interesting to use a similar approach as the one presented in the paper to evolve kinetic models that explicitly account for metabolite concentrations and isoenzymes. This is unlikely to be computationally feasible at genome-scale, but could shed light on isoenzyme evolution in focused models.*

Actions: *We now discuss the role of isozymes in the discussion on page 14 lines 289-294.*

2. On Line:154 the authors state that "small set of genes that was repeatedly responsible for large jumps in fitness". For this, they set all enzymes unconstrained resulting in the increase in growth rate. I was wondering if they tried simulating the model while keeping these enzymes/genes in the constrained set and compare the trajectories with the reported model.

Response: *We agree that this is a promising experiment to determine the effect of multifunctional enzymes on evolutionary dynamics. We thus removed reactions that are catalyzed by genes that were associated with jump dynamics (i.e., the list presented in Supplemental Table 1) from the evolving set and repeated the evolutionary simulations. We find that this removal of multifunctional enzymes strongly reduces jump behavior and shows an increased speed of adaptation, confirming our hypotheses on the role of multifunctional enzymes in k_{cat} evolution.*

Actions: *We added Supplemental Figure 6, report the results and statistical analysis on p.8 lines 160-163, and discuss them on page 13 lines 266-268.*

3. Line 267: It wasn't clear to me if the authors have included the effects of inhibitors and enhancers on the enzymes. To the best of my understanding, they haven't. If so, the authors might want to mention that they are ignoring these effects.

Response: We apologize for omitting this point; The Reviewer is correct, inhibitors and enhancers are not part of the current model.

Actions: We now discuss the effect of explicit modelling of metabolite and effector concentrations on page 14 lines 289-292.

Typos/minor errors

=====

1. Line # 119: Figure 2C does not exist in the manuscript
2. Figure S12 1: different terminologies used for the same reaction (biophysically and biochemically).
3. Line # 157: "(purH), and cell ..." should be "(purH), cell"
4. Line # 188: data set should be dataset
5. Line # 635: of the number of the number of (repetition)

Actions: We corrected these errors.

Reviewer #3 (Remarks to the Author):

This paper presents an interesting and innovative way of investigating the evolution of metabolism. The authors show how the effect of mutations that affect the catalytic rate of enzymes depends on the genetic background of the organism; their model enables them to identify changes in k_{cat} that have a strong beneficial effect (multifunctional and enzymes with a low current k_{cat}). It also becomes clear how a small number of biophysical constraints can determine the maximally achievable growth rate of a cell. Although the approach is interesting, some of the model assumptions can be questioned and in view of the assumptions, some of the model results are fairly trivial and would not need the brute-force approach of a Genome-Scale Model.

Model assumptions

All enzymes are fully saturated. The authors cite Bennet et al. (2009) to argue that this assumption is reasonable. It should however be noted that cells were grown on saturating amounts of nutrients in this study and that Bennet et al. did find that the enzymes in lower glycolysis were not saturated. Making this assumption and thereby discarding effects like product inhibition and allosteric effects, strongly simplifies the dependence of the growth rate on k_{cats} which is likely the reason that the authors find a smooth fitness landscape with only one maximum. What would happen if inequalities were to be used (i.e. $v \leq k_{cat} * e$), allowing for sub-saturation?

Response: We apologize for not giving sufficient detail on the role of enzyme sub-saturation. We agree that a more detailed kinetic model of metabolism that includes metabolite concentrations and enzyme affinities could improve predictive accuracy of k_{cat} evolution simulations. A major reason behind our choice to simulate proteome-limited metabolism with MOMENT is that it is a linear program. Typical simulations as those presented in Figure 2A require about 1000 CPU hours per replicate, even with the computationally efficient MOMENT algorithm. We thus think that a representation of more detailed non-linear kinetics in the current MCMC framework is limited by computational resources. Using a linear saturation term would circumvent this problem, but it would require knowledge about K_m and substrate concentrations at each intermediate evolutionary step, where K_m and k_{cat} frequently show trade-offs that need to be accounted for (e.g., Savir et al. 2010, PNAS 107, 3475-3480). Even in modern E.coli metabolism, no reaction can be adequately parameterized with K_m , substrate concentrations, and in vitro k_{cat} for validation (Davidi et al. 2016 PNAS), making reliable genome-scale parameterization infeasible, even if evolutionary growth conditions are assumed to be constant. We argue that the effects of sub-saturation are better tackled with smaller-than-genome scale kinetic models, where manual parameterization and efficient model solving is feasible. That being said, undersaturation increases protein costs of a reaction and is thus likely to be selected against. We can thus expect the bias in our predictions that is introduced by undersaturation to be mitigated by selection for high-affinity enzymes.

Actions: We now discuss the effect of undersaturation, K_m , and allosteric effects in more detail (page 14 lines 282-294) and explain how k_{cat} - K_m trade-offs could affect the shape of the fitness landscape (page 14 lines 301-303). We removed the reference to Bennet et al. (2009) to avoid misleading the reader with regard to the specific experimental conditions in that study.

Additionally, substrate specificity and turnover rates are constrained by thermodynamics of the catalyzed reaction via microscopic reversibility - the Haldane relationship. By assuming full saturation of all enzymes, this constraint is relieved, obviously, but there are quite a number of papers –e.g. from Ron Milo’s lab- showing the importance of thermodynamics in defining the costs associated with metabolic flux.

***Response:** We agree that backward flux in thermodynamically unfavorable reactions is indeed expected to exert additional selection pressure on k_{cat} , leading our model to underestimate the k_{cat} of strongly unfavorable reactions. A quantitative treatment of this effect would require a kinetic model that predicts steady-state metabolite concentrations at intermediate steps of evolution, which is currently computationally infeasible in our MCMC framework (as explained above).*

***Actions:** We discuss the effect of backward flux on k_{cat} evolution on page 14 lines 284-288. MDH is discussed as an example to illustrate the effect of thermodynamic constraints.*

The identity of biophysically constrained reactions is assumed known. I agree with the authors that their modelling framework is a nice starting point and that choosing the set of biophysically constrained reactions based on EC numbers is a good first approximation. However, we can’t be sure that enzymes that have low current k_{cats} are slow because of biophysical constraints or because of the very evolutionary history that the authors try to model. Using this set of constrained reactions therefore imposes part of the answer on the model. The authors also state that their results are robust against the identity of the constrained set. This is, in my opinion, only partly true:

In Supplementary Information 2 (e.g. Equation on line 636) the authors show that in a linear pathway the eventual growth rate is largely determined by the number of constrained reactions. Since it was mathematically proven (Wortel et al., Müller et al. (2014)) that resource allocation models such as MOMENT attain maximal growth rate in an Elementary Flux Mode (for which the authors’ derivation is equally valid as for a linear pathway), this result is true even in the genome-scale modelling case. It is therefore almost trivially true that the achieved growth rate is robust against the identity of the set, but not against the size of the set.

The actual important validation of the results - for which robustness should thus be tested- is the correlation of simulated k_{cats} and experimental k_{cats} . I suspect that this correlation is not robust at all against the identity of the set of constrained reactions.

***Response:** We agree that the identity of the constrained set is both of high interest and uncertainty. We thus extended our sensitivity analysis for the size and identity of the constrained set by repeating our analyses with random evolving sets of different sizes. We find that the correlation with experimental data is robust against the identity and size of the constrained set, supporting the idea that biomass-optimal flux and enzyme molecular weight are the main determinants of k_{cat} in the evolutionary end points. This notion is confirmed by the random perturbation of network stoichiometries and biomass components (see below for details).*

***Actions:** We added Supplemental Table 2 and Supplemental Figure 8 and referenced them in the discussion on page 14 lines 298-300. We further added statistical analysis to Supplemental Figure 7.*

Results

A smooth single-peaked phenotypic fitness landscape is found. This is a direct consequence of the model setup. The growth rate is a monotone function of all k_{cat} s. At the same time, there is a fundamental upper bound to the growth rate, derived in SI 2. Given enough iterations, all unbounded k_{cat} s will become as large as possible (although the rate of increase will slow down enormously). The Genome Scale Model is not needed for this result, and in fact masks the triviality of the outcome.

Response: *We agree that a lack of local fitness optima can be inferred from inspecting the model alone, but the model setup alone does not imply that evolution shows the high level of convergence that we observe. The fitness landscape does exhibit neutral plateaus that cause divergence (Supplemental Figure 1) and irregular adaptation dynamics (Figure 3). The magnitude of these neutral plateaus and the divergence they cause is thus not trivially predicted from the model setup alone.*

Actions: *We now discuss the effect that more complex kinetic models could have on the inferred shape of the fitness landscape on page 14 lines 301-303.*

The model predictions do correlate significantly with in vitro k_{cat} s. What I miss is an assessment of the robustness or triviality of this result (see also comment above). Would changes in the network stoichiometry, or even random networks, produce similar correlates? The correlation with in vitro data is (just) significant, but not necessarily impressive.

Response: *We agree that the agreement with experimental data is a central result of the paper and thus warrants extensive sensitivity analysis. We thus performed random perturbations of reaction stoichiometries and biomass components to estimate the effect of network structure on the predicted k_{cat} s and their correlation with experimental data. We find that these perturbations abolish the observed correlations, confirming the hypothesis that the natural network structure and the flux it carries are the major determinants of evolved k_{cat} s. Furthermore, we now assess the robustness of the correlation of modelled k_{cat} s with experimental data against average mutation sizes, constraint set size, constraint set identity, ancestral states, and the presence of evolving multifunctional enzymes. Significant correlations were found to be robust against all of these factors, although random ancestral states decrease correlations due to the scarcity of fixations that decrease k_{cat} s that were initialized to be high.*

Actions: *We added the results of random network perturbations in Supplemental Figure 14 and discuss them on page 15 lines 331-335. We further added sensitivity analyses of the correlation of predictions with experimental data against size and identity of the evolving set (Supplemental Table 2 and Supplemental Figure 8), jump size (added correlation to legend of Supplemental Figure 9), initial conditions (Supplemental Figure 12), and the presence of multifunctional enzymes (legend of Supplementary Figure 6).*

The model predictions are in better agreement with in vivo data than with in vitro data. The cause of this result is not entirely clear to me, and is also not extensively discussed or explained. To what extent can this be the result of the in vivo data being generated using an FBA-based approach too?

Response: *We think that the higher correlation of in vivo data in comparison to in vitro data is due to a higher noise level in in vitro data (see the SI of Bar Even et al 2011 Biochemistry, 2011, 50, 4402-4410 for discussion of this issue) and a bias towards in silico-derived fluxes, as we state in the discussion. That being said, although little experimental data on fluxes across diverse conditions is available, $k_{app,max}$*

derived from MFA-estimates was shown to have a high correlation with estimates derived from in silico fluxes ($R^2=0.85$, Davidi et al. PNAS 2016, 113, 3401-3406), indicating that the bias introduced by using in silico fluxes is low. Unfortunately the raw data underlying this analysis was to our knowledge not published, so we could not check for agreement with our predictions.

Actions: *We extended our discussion of the superior performance of our model on in vivo data on page 16 lines 339-345.*

“The evolutionary long-term end points of kcat evolution can be predicted using systems models with unexpected accuracy”. I don’t think such a strong statement is validated when a non-negligible set of kcats is imposed on the model, especially because fixing these parameters strongly influences the predicted endpoints.

Response: *We agree that “unexpected accuracy” is an unnecessarily subjective claim. We rephrased the statement to make it more objective.*

Actions: *We rephrased the statement on page 16 lines 347-350.*

Minor typo’s

- The growth rate on line 130 should be per hour, not per second

Actions: *Corrected the error.*

Reviewer #4 (Remarks to the Author):

This paper outlines an excellent, and very timely approach to simulating metabolic evolution. While the findings are entirely consistent with MCA theory, nobody has ever applied such a broad, grand approach to networks. A tour de force, a long string of excellent points were made, and the writing is quite good, too.

My only real “objection” is with the anti-MCA tone that comes across, as by my reading there is broad similarity in approaches. Consider, for example, the idea that changes in a single enzyme will exhibit diminishing returns. Although you are changing many enzymes, the intuition is essentially the same: the more you fix the fixable problems, the less they are a problem (i.e., their control coefficients go down). Your approach is phenomenally more sophisticated in breadth, and doesn’t rely upon linearization around local changes to the steady-state – so it is distinctly more realistic – but the root cause is not so different. And you should occasionally have seen synergistic (positive) epistasis for single-function enzymes if you have two in a row that hit the same pathway. The first such mutation in enzyme A will decrease its control coefficient but raise that for enzyme B...

Response: It was not our intention to criticize MCA in any way, we were rather trying to make the conceptual differences clear to the reader. We agree that, if a single reaction was part of the evolving set, the situation would be the same as those discussed in the MCA literature.

Actions: We rephrased our comparison to MCA on page 12 lines 238-239.

Was it really necessary to assume only 1% of mutations increased k_{cat} if you are modeling adaptation under SSWM? Did you ever find BENEFICIAL drops in activity, for example? I know this is possible due to ultrasensitivity around branchpoints, but I think it deserves comment.

Response: We decided to use a low rate of beneficial mutations because of the nature of active site optimization, where the majority of random changes have to be assumed to decrease k_{cat} . In addition, deleterious mutation will still have fixation probabilities greater zero in the population genetics model, and we indeed observe a few deleterious steps per replicate in the trajectories presented in Figure 2A. Although we agree that our general results would not change if we ignored the high rate of deleterious mutations, we still think that it renders the model set-up closer to reality.

Actions: We now discuss the role of deleterious mutations on pages 14-15 lines 307-317.

The jumps in fitness are quite fun, but I feel that there is a second reason (besides the VERY well-explained constraint brought up) that this can occur in a model but not in an experiment: SSWM versus clonal interference. If you actually had a population making mutations that mimicked the N_e of a flask, and fixations were not instantaneous, then clonal interference between lineages should cause the greatest gains to occur early on rather than later. It would be worth mentioning this aspect, I think. (And maybe it becomes a future paper comparing the difference.)

Response: We agree that clonal interference could affect the dynamics of k_{cat} evolution, especially in large microbial populations. This is especially true for potential future studies of the time-scale of k_{cat} evolution.

Actions: *We now discuss the effect of clonal interference on page 14 lines 318-322.*

At first read in the abstract on predicting experimental k_{cat} seemed to make a big claim (that I was quite dubious about). When I got down to that section, I understood the extent of the claim, but it would be useful from the start to say that X portion of the variance was explained. What still isn't spelled out is why the hell it should work at all! My guess is that the predicted flux values needed for optimal growth create the primary metabolism/secondary metabolism sort of scenario that leads to the need for higher k_{cat} in high flux reactions. Is this indeed the case? Just a bit more explanation about this would be quite helpful, or perhaps show that a distinct assumption – like growth on something like acetate – would lead to a poorer correlation because it would underestimate glycolysis, etc.

Response: *We agree that the prediction of experimental data is a central result of the paper that deserves more discussion. The high correlations between average flux across conditions and simulated endpoint k_{cat} s indicates that the wide spread of flux magnitudes indeed leads to differential selection pressures that are—in combination with molecular enzyme weight—an important determinant of end point k_{cat} s. This idea is confirmed when network stoichiometries and biomass components are perturbed randomly: the correlation with experimental data is abolished (Supplemental Figure 14).*

Actions: *We now added Pearson's correlation coefficients for predictions and data to the abstract (page 1 line 22). Furthermore, we now discuss how a large fraction of the variance in simulated k_{cat} s is explained by differences in average flux and molecular weight, and how random perturbation of network stoichiometries and biomass components confirms the importance of network structure and flux (page 15 lines 327-335).*

Minor comment:

p.5 – Out of curiosity, as it was hilariously ridiculous, what constrains the “unconstrained” case to a doubling time of 1.58 per second? Is that just some computational thing?

Response: *This was simply a typographical error; we meant “h⁻¹”, and apologize for any confusion this might have caused.*

Actions: *Corrected error.*

Reviewers' comments:

Reviewer #1 (Remarks to the Author):

The authors addressed all my concerns. I now recommend publication of their paper in Nature Communications.

Reviewer #2 (Remarks to the Author):

Thank for your making the corrections/changes.

Reviewer #3 (Remarks to the Author):

[See page 21 of this Peer Review File].

Reviewer #4 (Remarks to the Author):

The authors have done a commendable job answering my comments, as well as those of the other reviewers. I have no further concerns.

Reviewer #3 (Remarks to the Author):

Overall the authors did revise the paper according to my comments, and so they weaken some of the claims and explain some of the assumptions and limitations, and they do some additional sensitivity analyses to substantiate some of the statements. But I still have problems with the manuscript. The work is rather unsatisfying as it presents the results as observations, without a good theoretical or computational underpinning. The latter is the reason why models can be so helpful, and why simplifying assumptions are sometimes warranted. Without such deeper analysis, this study does not help in understanding, for example, the question posed at the beginning: why primary metabolism has higher k_{cat} s than secondary metabolism. Let me therefore explain - better than I apparently did- why I think part of the work is flawed, part is trivial, and part is preliminary.

Where it is flawed:

Enzyme-catalyzed reactions have to obey the laws of thermodynamics. This means that for a Michaelis-Menten type reaction, when the mass action ratio of products and substrates equals the equilibrium constant, the net rate is zero (as the system is in equilibrium, then). From this it follows, for a one substrate, one product reaction, that $\frac{[P]_{\text{eq}}}{[S]_{\text{eq}}} \equiv K_{\text{eq}} = \frac{k_{\text{cat}}^+ K_s}{k_{\text{cat}}^- K_p}$. Here K_s is the K_m of the substrate, etc. This is the Haldane relationship I referred to. So it is impossible to change only the k_{cat} (of the forward direction); any mutation that affects k_{cat} also affects another parameter. This creates tradeoffs that may also -or additionally- explain why k_{cat} s are not as high as theoretically possible: it would require a poor K_m or a high backward rate. Now, the answer given by the authors as a response why this is ignored, is that they cannot do this for large networks because we do not know the kinetics. Because they cannot do it, does not mean they can ignore it! By the way, I think it is possible (an interesting) to look at the evolution of k_{cat} s in full kinetic complexity, indeed probably at a small scale, for a specific pathway, but people do kinetic genome-scale models, and with sampling approaches and approximative kinetic formats relevant insight on the importance of thermodynamic constraints could be obtained. So I find it a bit too easy to simply ignore such a potential confounding factor.

Where it is trivial:

As explained well in the supplementary materials, the DRE (diminishing returns) is a simple consequence of the model setup: if there are two enzymes, and one of them is constrained with a fixed $k_{\text{cat},c}$, and the total amount of enzyme (C) is constrained, and it is dictated that $v = k_{\text{cat}} [E]$, then the maximal rate through the pathway is completely fixed and predetermined: the maximal flux is then simply $k_{\text{cat},c} C$ when the k_{cat} of the unconstrained enzyme approaches infinity. Extending this to a genome scale model is trivial: given a maximal growth rate (determined by the fixed k_{cat} s and C) and the type of constrained optimisation problem at hand, the optimal flux solution is also fixed and known as an Elementary Flux Mode. This means that for each step in the network the rate is known. Now, it is just a matter of optimally filling up the protein space according to the required fluxes and the MW of the proteins. There is most likely a unique optimal solution, and that could explain the smoothness of the fitness landscape and why the evolutionary trajectories are so convergent. Why is this not explained properly, with the proper literature cited?

Where it is preliminary:

What is not fully explained and not so trivial, it seems, is that the trajectories apparently converge to a suboptimal solution, at a growth rate of 0.5 h^{-1} where apparently 1.58 h^{-1} could be reached if all k_{cat} s are simultaneously increased, not one by one. This suboptimality is explained by DRE. This is of course very interesting and would be the key message of the paper, but it is not examined much deeper. Ok, DRE explains why you may not reach the optimal state, because drift takes over at a certain point. But why do they converge, and why to 0.5 h^{-1} , and why to similar k_{cat} distributions? This must hint to some other constraints? What if mutation frequencies or other probabilities are changed, or the effective population size? There is not even an attempt to explain this.

The work is thus very unsatisfying; it is telling for the lack of thoroughness, that the question about high k_{cat} s for primary metabolism is not revisited and answered, even though the answer may be found in the results. Let me try to do some of the thinking, then: If all the fluxes are fixed through an EFM, then the highest fitness gain is achieved by a fractional increase in the k_{cat} of the highest flux reaction - weighted by the costs in terms of MW. These high fluxes will be in primary metabolism. Now, it is of course interesting that the in silico k_{cat} s correlate (somewhat) with in vitro k_{cat} s. It shows that perhaps protein costs are indeed drivers of selection; not necessarily a new concept, but an interesting indirect experimental test, perhaps. It probably "just" means that for E coli we have the stoichiometries and MWs right: scrambling stoichiometry and scrambling MW will affect the filling up according to flux and MW and will thus destroy the correlation, as the authors showed in the revised manuscript. It also implies that there is a correlation with MW and k_{cat} , as there is with flux, something they can test in the data.

But first they need to understand the convergence to a suboptimal endpoint of the evolutionary trajectories. Without that, there is no understanding, and then, according to me, there is not much left.

Reviewer comments with responses and actions taken

Comments by Reviewer 3

Overall the authors did revise the paper according to my comments, and so they weaken some of the claims and explain some of the assumptions and limitations, and they do some additional sensitivity analyses to substantiate some of the statements. But I still have problems with the manuscript. The work is rather unsatisfying as it presents the results as observations, without a good theoretical or computational underpinning. The latter is the reason why models can be so helpful, and why simplifying assumptions are sometimes warranted. Without such deeper analysis, this study does not help in understanding, for example, the question posed at the beginning: why primary metabolism has higher k_{cat} s than secondary metabolism. Let me therefore explain - better than I apparently did- why I think part of the work is flawed, part is trivial, and part is preliminary.

Where it is flawed:

Enzyme-catalyzed reactions have to obey the laws of thermodynamics. This means that for a Michaelis-Menten type reaction, when the mass action ratio of products and substrates equals the equilibrium constant, the net rate is zero (as the system is in equilibrium, then). From this it follows, for a one substrate, one product reaction, that $P_{eq} / S_{eq} = K_{eq} = (k_{cat+} * K_s) / (k_{cat-} * K_p)$. Here K_s is the K_m of the substrate, etc. This is the Haldane relationship I referred to. So it is impossible to change only the k_{cat} (of the forward direction); any mutation that affects k_{cat} also affects another parameter. This creates tradeoffs that may also -or additionally- explain why k_{cat} s are not as high as theoretically possible: it would require a poor K_m or a high backward rate. Now, the answer given by the authors as a response why this is ignored, is that they cannot do this for large networks because we do not know the kinetics. Because they cannot do it, does not mean they can ignore it!

By the way, I think it is possible (an interesting) to look at the evolution of k_{cat} s in full kinetic complexity, indeed probably at a small scale, for a specific pathway, but people do kinetic genome-scale models, and with sampling approaches and approximative kinetic formats relevant insight on the importance of thermodynamic constraints could be obtained. So I find it a bit too easy to simply ignore such a potential confounding factor.

Response: We apologize for not clearly stating the way we account for the Haldane relationship, leading to the impression that we ignored the latter. In our simulation algorithm, the k_{cat} s for the forward and the reverse direction of reversible reactions are changed simultaneously and by the same alpha value. Thus, the simulated mutations respect the Haldane relationship and do not change the K_m s of the enzyme. Such mutations can increase the flux capacity of an enzyme without changing the relative amount of backward flux that would result for a given set of metabolite concentrations; this can be seen by decomposing reversible Michaelis Menten kinetics (e.g., Equation 4 in Noor et al. 2016, PLoS Comput Biol 12, 1-29). As we discuss in the manuscript, a quantification of the cost of backward flux as a function of variable metabolite concentrations requires a more detailed kinetic model that is not in the scope of this study and—as the reviewer points out—would be more applicable to small scale study.

Action taken: We now explain our treatment of k_{cat} s for the reverse direction in accordance with the Haldane relation in the Methods section (l.430-432).

Where it is trivial:

As explained well in the supplementary materials, the DRE (diminishing returns) is a simple consequence of the model setup: if there are two enzymes, and one of them is constrained with a fixed $k_{cat,c}$, and the total amount of enzyme (C) is constrained, and it is dictated that $v = k_{cat} [E]$, then the maximal rate through the pathway is completely fixed and predetermined: the maximal flux is then simply $k_{cat,c} C$ when the k_{cat} of the unconstrained enzyme approaches infinity. Extending this to a genome scale model is trivial: given a maximal growth rate (determined by the fixed k_{cats} and C) and the type of constrained optimisation problem at hand, the optimal flux solution is also fixed and known as an Elementary Flux Mode. This means that for each step in the network the rate is known. Now, it is just a matter of optimally filling up the protein space according to the required fluxes and the MW of the proteins. There is most likely a unique optimal solution, and that could explain the smoothness of the fitness landscape and why the evolutionary trajectories are so convergent. Why is this not explained properly, with the proper literature cited?

Response: The idea that a single elementary flux mode (EFM) determines the direction of evolutionary trajectories throughout the adaptation process and thus explains the observed convergence of k_{cat} evolution is appealing. Nevertheless, we find the same level of convergence in the scenario where we randomly change growth environments throughout evolution. Surely this behavior cannot be explained by the usage of a scalar multiple of a single EFM, and the convergent behavior is thus not trivial. Instead, our explanation of the observed convergence is the following: We show that reaction flux and enzyme molecular weight are major determinants of evolved k_{cats} in the end points of our simulations, where flux has a stronger effect than molecular weight. Even though flux distributions change in response to the current environment, flux distributions are correlated across environments: e.g., many sugars are catabolized through glycolysis. We confirm this notion by examining the correlation matrix of flux distributions predicted by MOMENT across 10,000 growth conditions and find a median Pearson's R of 0.7. As enzyme molecular weight is constant through the simulation, selection pressure on a given enzyme is correlated across growth conditions, causing the observed convergent behavior of k_{cat} evolution. Thus, contrary to the notion that a single EFM explains convergent behavior, we find that diverse flux distributions result in convergence. This finding is a non-trivial conclusion resulting from the structure of the metabolic network and quantitative proteome requirements for growth across diverse conditions.

Action taken: We now highlight the effect of flux and enzyme molecular weight on selection pressure (l.210-218) and added the convergence across diverse conditions to the results section (l.219-221). Furthermore, we discuss the effect of correlated flux distributions as the cause for convergent behavior (l.271-276).

Where it is preliminary:

What is not fully explained and not so trivial, it seems, is that the trajectories apparently converge to a suboptimal solution, at a growth rate of 0.5 h⁻¹ where apparently 1.58 h⁻¹ could be reached if all k_{cats} are simultaneously increased, not one by one. This suboptimality is explained by DRE. This is of course very interesting and would be the key message of the paper, but it is not examined much deeper. Ok, DRE explains why you may not reach the optimal state, because drift takes over at a certain point. But

why do they converge, and why to $0.5 h^{-1}$, and why to similar k_{cat} distributions? This must hint to some other constraints? What if mutation frequencies or other probabilities are changed, or the effective population size? There is not even an attempt to explain this.

Response:

“why do they converge” (to a sub-optimal solution)?

We agree that the high distance of our end points from the theoretical optimum is an important point of the paper that warrants further investigation. To understand this behavior, we must understand what factors drive the selection coefficient s of mutations that increase k_{cat} to values that are close to $1/N_e$ (where N_e is the effective population size), even when the system is still far from a global optimum. To this end, we now extended the simple analytical model we present in SI2 to quantify the effect of different system parameters on the selection coefficient s . We find that the number of reactions in the system M is a major determinant of the selection coefficient s that is associated with a mutation that changes a single k_{cat} . In the example of the linear pathway, a system on the order of one thousand reactions can reach very low selection coefficients, even though it is still far from the theoretical optimum.

Action taken: We extended SI2 with an analysis of factors that affect the selection coefficient s (l. 197-217 of the SI) and add our findings about the effect of network size to the results section (l. 132-133) and the discussion (l. 366-371 & l. 375-376).

“and why to similar k_{cat} distributions”?

For the system properties that affect convergence please see our response and action taken in regard to your second point (“Where it is trivial”).

“and why to $0.5 h^{-1}$ ”?

To further elucidate why the final growth rate settles around $0.5 h^{-1}$, we conducted a sensitivity analysis against the magnitude of the k_{cat} of biochemically constrained reactions. As expected, we find that the magnitude of these constraints has a major impact of the final growth rate. For a sensitivity analysis against the probability distribution of mutation effects see Supplementary Figure 9.

Action taken: We add the results of the new sensitivity analysis against constrained k_{cat} s in Supplementary Figure 10.

The work is thus very unsatisfying; it is telling for the lack of thoroughness, that the question about high k_{cat} s for primary metabolism is not revisited and answered, even though the answer may be found in the results. Let me try to do some of the thinking, then: If all the fluxes are fixed through an EFM, then the highest fitness gain is achieved by a fractional increase in the k_{cat} of the highest flux reaction - weighted by the costs in terms of MW. These high fluxes will be in primary metabolism. Now, it is of course interesting that the in silico k_{cat} s correlate (somewhat) with in vitro k_{cat} s. It shows that perhaps protein costs are indeed drivers of selection; not necessarily a new concept, but an interesting indirect experimental test, perhaps. It probably “just” means that for E coli we have the stoichiometries and MWs right: scrambling stoichiometry and scrambling MW will affect the filling up according to flux and MW and will thus destroy the correlation, as the authors showed in the revised manuscript. It also

implies that there is a correlation with MW and k_{cat} , as there is with flux, something they can test in the data.

Response: We apologize that the answer to the central question of the manuscript, the occurrence of high k_{cat} s in primary metabolism, was not clearly presented.

The correlation between flux and enzyme molecular weight in endpoint k_{cat} s shows how the speed of evolution is higher in k_{cat} s of large enzymes that catalyze high flux reactions. As beneficial mutations accumulate throughout adaptation, DRE gradually reduces fitness gains until the neutral barrier is reached and improvements in any k_{cat} are close to neutral. We expect the highest fluxes to occur in primary metabolism, thus enzymes in primary metabolism will hold the highest k_{cat} s in the evolutionary end points. We have now extended the discussion with an additional paragraph that summarizes our theory of k_{cat} evolution as it emerges from the results in the paper. Furthermore, we now place the correlation between flux and simulated endpoint k_{cat} s more prominently in the results section in addition to its mention in the discussion section.

Actions taken: Added novel paragraph to the Discussion (l. 364-376) and mentioned the connection between flux and primary metabolism in the Results section (l. 214-215).

But first they need to understand the convergence to a suboptimal endpoint of the evolutionary trajectories. Without that, there is no understanding, and then, according to me, there is not much left.

Response:

Please see our response to section “Where it is preliminary”.

Additional changes:

- Added statements on code and data availability
- The legend of Supplementary Figure 1 now correctly reads “Pearson’s R” and not “R²”. This change does not affect any of our conclusions on the convergence of k_{cat} evolution.
- Made the spelling of “end point” consistent throughout the manuscript.
- Shortened abstract to meet formatting guidelines
- Formatted equations according to formatting guidelines
- Shortened headings in the Results section to meet formatting guidelines
- Removed headings in Discussion to meet formatting guidelines
- “Supplemental” prefix is now “Supplementary” to meet formatting guidelines (this change is not marked for the sake of facility of inspection)
- Referenced all supplementary figures.

Reviewers' comments:

Reviewer #3 (Remarks to the Author):

Actual results and the corresponding missed opportunities

Diminishing Returns Epistasis (DRE) could prevent *E. coli* from reaching its theoretical fitness optimum. This is an interesting result, however, this effect was already known. In fact, the unreferenced paper by Martin et al. (Nature Genetics 2007) investigated this effect in much more detail and generality. The authors could have further investigated what model parameters influence DRE. For example, the authors assume a 100:1 ratio of deleterious vs advantageous mutations. This ratio will affect much of the model predictions, and probably reducing the fit with experimental data.

The toy model in Supplementary Note 1 explains why there is a correlation between *k_{cat}*, Molecular Weight, and reaction rate.

This toy model is valid for all pathways with only one flux degree of freedom. We call these pathways Elementary Flux Modes (EFMs). The MOMENT-optimization that the authors use will always select such an EFM (this is a mathematical fact proven in papers that were again unreferenced, even after our rather explicit hint about EFMs in the previous report: Wortel et al., Müller et al. (2014)), and the toy model is thus always valid. It does not matter that the authors use changing conditions, this will only lead to a different EFM to be selected. This toy model, although perhaps less appealing because of its simplicity, is therefore the main result of the paper, rather than the complicated evolutionary algorithm that the authors use.

Trivial results

The correlation of predicted *k_{cat}*s with experimentally measured *k_{cat}*s.

Although this result will probably appeal to the general public, it is in fact a very trivial fit with a model that is needlessly complex. This is how the model works:

1. The reactions are split in 1087 constrained reactions and 569 unconstrained reactions. The *k_{cat}*s of the constrained reactions are fixed to a pre-specified value.
2. The unconstrained *k_{cat}*s are evolved, and thus increase.
3. The unconstrained *k_{cat}*s stop increasing when the growth rate is mostly limited by the constrained reactions.

The evolutionary algorithm will always lead to high *k_{cat}*s for unconstrained reactions, which will not give a good fit to experimental data for reactions with a low *k_{cat}*. What the authors have done is leave out all these potentially problematic reactions by marking them as the constrained reactions. In short, they try to predict 1656 *k_{cat}*s, throw out 1087 problematic ones, and then happily present the fit of the other 569 reactions.

It would have been very interesting if the authors had identified a mechanistic characteristic of biophysically constrained reactions, e.g. a thermodynamic property. If they had picked the constrained set based on this characteristic, then a correct prediction would have been very impressive. However, the authors have not added any such mechanistic explanation in their revision, although we made this comment already in our first referee report:

"However, we can't be sure that enzymes that have low current *k_{cat}*s are slow because of biophysical constraints or because of the very evolutionary history that the authors try to model. Using this set of constrained reactions therefore imposes part of the answer on the model."

Comments of Reviewer 3 with detailed responses

Actual results and the corresponding missed opportunities

Diminishing Returns Epistasis (DRE) could prevent *E. coli* from reaching its theoretical fitness optimum. This is an interesting result, however, this effect was already known. In fact, the unreferenced paper by Martin et al. (Nature Genetics 2007) investigated this effect in much more detail and generality.

Response: We are aware of the important work of Martin et al. (2007), but the scope of their paper is fundamentally different from the problems approached in our study.

Martin et al. (2007) use an extension of Fisher's geometric model to investigate distributions of epistasis that arise from stabilizing selection on a single-peaked landscape when the phenotype is in proximity to the optimum (see Martin et al. (2006) for a discussion of the scope of the model used in Martin et al. (2007)). The Reviewer states that the fact that *E. coli* is prevented from reaching the fitness optimum is an interesting result of our study, but that Martin et al. (2007) stated this before us. It is the quantitative *distance* from the optimum that constitutes the interesting result, and Martin et al. (2007) make no statement about this distance. Quantifying the distance from the optimum requires the kind of genome-scale approach that we are taking here, as this distance will be a function of the number of fitness-relevant enzymes—which is in turn a function of the environment—and the effective population size that is specific for *E. coli*. Furthermore, the model presented in Martin et al. (2007) would not be applicable to the specific scenario of k_{cat} evolution because of three key points: 1) in our theoretical framework, k_{cat} evolution does not take place on a single-peaked fitness landscape and is **not subject to stabilizing selection**. 2) we model **diverse and changing environments**, a scenario for which Fisher's geometric model can not be readily applied (see Martin et al. (2006), Orr (2005) for discussion) 3) we model macroevolution. The population is **far from the population optimum** in the ancestral state and, as we show through our simulations, remains far from the optimum due to population size and system size.

In fact, Martin et al. (2007) see their approach as an “alternative” to the metabolic model based approach by Segre et al. (2005)—a work that is conceptually closer to our approach and that we do cite—for situations where detailed mechanistic models are unavailable.

In summary, Martin et al. (2007) did not seek to answer the questions we address in the manuscript, and, even if they had, they could not have applied their model to the scenario we are modelling.

The authors could have further investigated what model parameters influence DRE. For example, the authors assume a 100:1 ratio of deleterious vs advantageous mutations. This ratio will affect much of the model predictions, and probably reducing the fit with experimental data.

Response: We conducted a sensitivity analysis against the ratio of deleterious mutations, varying it across three orders of magnitude. We find that the model's predictions and ability to

explain experimental data are robust to changes in the ratio of deleterious to advantageous mutation rates. The underlying reason for this behavior is the strong selection pressure (i.e., low fixation probabilities) against deleterious mutations, allowing only deleterious mutations that are close to neutral to be fixed at meaningful rates, even when deleterious mutation rates are very high.

Actions: We added the results of the new analysis as Supplementary Table 3.

The toy model in Supplementary Note 1 explains why there is a correlation between k_{cat} , Molecular Weight, and reaction rate.

This toy model is valid for all pathways with only one flux degree of freedom. We call these pathways Elementary Flux Modes (EFMs). The MOMENT-optimization that the authors use will always select such an EFM (this is a mathematical fact proven in papers that were again unreferenced, even after our rather explicit hint about EFMs in the previous report: Wortel et al., Müller et al. (2014)), and the toy model is thus always valid. It does not matter that the authors use changing conditions, this will only lead to a different EFM to be selected. This toy model, although perhaps less appealing because of its simplicity, is therefore the main result of the paper, rather than the complicated evolutionary algorithm that the authors use.

Response: This argument is built on the assumption that the analytical model in Supplementary Note 1 “explains why there is a correlation between k_{cat} , Molecular Weight, and reaction rate”. This statement is not true, the analytical model does not explain this correlation. Even if we had an analytical model that explained the correlation, the validation with experimental data that we conducted for the genome-scale simulations—the central result of our study—would not be possible with a toy model. It is an interesting result that MOMENT yields flux distributions that follow EFMs, but which EFM is chosen at any particular time point in the evolutionary process will be a function of the current k_{cat} vector and the current environment. Thus, the fact that MOMENT chooses EFMs is an interesting technical detail, but it does not allow the simple analytical model to replace the complex adaptation behavior captured by our simulations. The general convergent behavior we find in our simulations is one example; the divergence of some reactions (Supplementary Figure 1) shows that this result is far from trivial. The global convergence we find suggests that the end points of k_{cat} evolution are predictable, allowing the experimental validation of our model (Figure 4). A thorough validation with experimental data requires the predictions of end point k_{cat} s that arise from our simulations and would not have been possible with a simple analytical model. In summary, the **central results of our paper require the genome-scale simulations** that we conducted and **could not have been achieved with the simple analytical model** presented in Supplementary Note 1.

Trivial results

The correlation of predicted k_{cats} with experimentally measured k_{cats} .

Although this result will probably appeal to the general public, it is in fact a very trivial fit with a model that is needlessly complex. This is how the model works:

1. The reactions are split in 1087 constrained reactions and 569 unconstrained reactions. The k_{cat} s of the constrained reactions are fixed to a pre-specified value.
2. The unconstrained k_{cat} s are evolved, and thus increase.
3. The unconstrained k_{cat} s stop increasing when the growth rate is mostly limited by the constrained reactions.

The evolutionary algorithm will always lead to high k_{cat} s for unconstrained reactions, which will not give a good fit to experimental data for reactions with a low k_{cat} . What the authors have done is leave out all these potentially problematic reactions by marking them as the constrained reactions. In short, they try to predict 1656 k_{cat} s, throw out 1087 problematic ones, and then happily present the fit of the other 569 reactions.

Response: We do not see how the Reviewer could arrive at the conclusion that “What the authors have done is leave out all these potentially problematic reactions by marking them as the constrained reactions”: As we state in the Methods section, we do not apply constraints to any of the reactions that have data available, precisely to avoid the problem the Reviewer is pointing out here. Furthermore, we further investigated this exact problem in the first round of revisions in response to comments brought up by Reviewer 3 and Reviewer 1: Supplemental Table 2 shows that randomly chosen sets of constrained and evolving reactions result in significant correlations with experimental data for both k_{cat} *in vitro* and $k_{app,max}$ that are comparable or higher than those shown for the original constraint set (see Supplementary Figure 11). The model predictions are thus highly non-trivial.

It would have been very interesting if the authors had identified a mechanistic characteristic of biophysically constrained reactions, e.g. a thermodynamic property. If they had picked the constrained set based on this characteristic, then a correct prediction would have been very impressive. However, the authors have not added any such mechanistic explanation in their revision, although we made this comment already in our first referee report:

“However, we can’t be sure that enzymes that have low current k_{cat} s are slow because of biophysical constraints or because of the very evolutionary history that the authors try to model. Using this set of constrained reactions therefore imposes part of the answer on the model.”

Response: This criticism is predicated on the idea brought up above, namely that “Using this set of constrained reactions therefore imposes part of the answer on the model”. The manuscript contains a sensitivity analysis (Supplementary Table 2) that shows that the choice of constraints has little influence on the ability of our model to explain experimental data. Thus, as we discuss in the manuscript, we cannot identify the exact identity of the constrained set as of yet, but this fact does not affect the validity of our results.

References

- Martin, Guillaume, Santiago F. Elena, and Thomas Lenormand. 2007. “Distributions of Epistasis in Microbes Fit Predictions from a Fitness Landscape Model.” *Nature Genetics* 39 (4): 555–60.

Martin, Guillaume, and Thomas Lenormand. 2006. "A General Multivariate Extension of Fisher's Geometrical Model and the Distribution of Mutation Fitness Effects across Species." *Evolution; International Journal of Organic Evolution* 60 (5): 893–907.

Orr, H. Allen. 2005. "Theories of Adaptation: What They Do and Don't Say." *Genetica* 123 (1-2): 3–13.

Segre, Daniel, Alexander DeLuna, George M. Church, and Roy Kishony. 2005. "Modular Epistasis in Yeast Metabolism." *Nature Genetics* 37 (1): 77–83.

REVIEWERS' COMMENTS:

Reviewer #5 (Remarks to the Author):

The paper seems to be based on solid work, and to have addressed all questions on sensitivity to unknown parameters that one would want to see in a paper of this type.

The only aspect of the response to the reviewer's first comments that I found puzzling is the resistance to acknowledging other work that has addressed the question of diminishing epistasis and fitness from different perspectives. I don't think that the Martin 2007 paper makes the current paper less interesting, but feel it is a pertinent work to quote. In fact, the manuscript doesn't cite or acknowledge in the introduction other classical and very pertinent work on diminishing return epistasis, such as the work by Lenski and colleagues. Again, these are papers that identified this phenomenon using different approaches and models. The current paper is unique in its mechanistic evaluation and insight, so it shouldn't be a problem to place it in the context of these other works, and I think it would in fact be beneficial for the authors.

Reviewer comments with responses and actions taken

Reviewer #5 (Remarks to the Author):

The paper seems to be based on solid work, and to have addressed all questions on sensitivity to unknown parameters that one would want to see in a paper of this type.

The only aspect of the response to the reviewer's first comments that I found puzzling is the resistance to acknowledging other work that has addressed the question of diminishing epistasis and fitness from different perspectives. I don't think that the Martin 2007 paper makes the current paper less interesting, but feel it is a pertinent work to quote. In fact, the manuscript doesn't cite or acknowledge in the introduction other classical and very pertinent work on diminishing return epistasis, such as the work by Lenski and colleagues. Again, these are papers that identified this phenomenon using different approaches and models. The current paper is unique in its mechanistic evaluation and insight, so it shouldn't be a problem to place it in the context of these other works, and I think it would in fact be beneficial for the authors.

Response: We thank the Reviewer for their thoughts on placing our work in the relevant literature. We were initially reluctant to cite Martin et al. 2007 because they focus on patterns of epistasis when diminishing returns are assumed *a priori* in the form of a Gaussian fitness landscape. This *a priori* assumption is made by a plethora of theoretical studies, but the mechanistic understanding of the cause for diminishing returns is lacking behind. That being said, Martin et al. 2007 serves as a good example of a study that assumes diminishing returns *a priori* and we thus now cite it in that context.

Lenski and colleagues have indeed made important experimental contributions to the nature of diminishing returns. We apologize for not citing their work before and now added a discussion of their results from the long-term evolutionary experiment.

Actions: Added citations of Martin et al. (Introduction and Discussion, l.35 and l.218-219) and of Khan et al. (Discussion l.220-222).

Additional changes

- Made Figure 1, 2, 3, and 4 more friendly for the color-blind.
- Added result summary to the end of the introduction.
- Made some sentences more accessible by giving more explanation (l.200-201, l.242, l.292, l.336).
- Made wording more precise when referring to *in vitro* kcats (“turnover number” instead of “turnover rate”).